# Recognition and control of neutrophil extracellular trap formation by MICL

Mariano Malamud[1], Lauren Whitehead[2], Alasdair McIntosh[3], Fabio Colella[4], Anke J. Roelofs[4], Takato Kusakabe[5,6], Ivy M. Dambuza[1,2], Annie Phillips-Brookes[1], Fabián Salazar[1], Federico Perez[7], Romey Shoesmith[1], Przemyslaw Zakrzewski[8], Emily A. Sey[1], Cecilia Rodrigues[1], Petruta L. Morvay[1,2], Pierre Redelinghuys[2], Tina Bedekovic[1], Maria J. G. Fernandes[9], Ruqayyah Almizraq[10], Donald R. Branch[10], Borko Amulic[8], Jamie Harvey[1], Diane Stewart[2], Raif Yuecel[11], Delyth M. Reid[2], Alex McConnachie[3], Matthew C. Pickering[12], Marina Botto[12], Iliyan D. Iliev[5,6], Iain B. McInnes[3], Cosimo De Bari[4], Janet A. Willment[1,2] & Gordon D. Brown[1,2✉]

Regulation of neutrophil activation is critical for disease control. Neutrophil extracellular traps (NETs), which are web-like structures composed of DNA and neutrophil-derived proteins, are formed following pro-inflammatory signals; however, if this process is uncontrolled, NETs contribute to disease pathogenesis, exacerbating inflammation and host tissue damage[1,2]. Here we show that myeloid inhibitory C-type lectin-like (MICL), an inhibitory C-type lectin receptor, directly recognizes DNA in NETs; this interaction is vital to regulate neutrophil activation. Loss or inhibition of MICL functionality leads to uncontrolled NET formation through the ROS–PAD4 pathway and the development of an auto-inflammatory feedback loop. We show that in the context of rheumatoid arthritis, such dysregulation leads to exacerbated pathology in both mouse models and in human patients, where autoantibodies to MICL inhibit key functions of this receptor. Of note, we also detect similarly inhibitory anti-MICL autoantibodies in patients with other diseases linked to aberrant NET formation, including lupus and severe COVID-19. By contrast, dysregulation of NET release is protective during systemic infection with the fungal pathogen *Aspergillus fumigatus*. Together, we show that the recognition of NETs by MICL represents a fundamental autoregulatory pathway that controls neutrophil activity and NET formation.

The immune system needs to balance immune responses to be able to control infectious challenge but at the same time avoid excessive inflammation that could generate host tissue damage[3,4]. Neutrophils, the most abundant immune cell type in circulation, contribute to the first line of defence against a large number of pathogens and are critical in this fine balance as their antimicrobial responses must be tightly regulated to maintain homeostasis[5,6]. Neutrophils are activated based on pro-inflammatory signals that trigger various effector functions depending on their surface receptor composition and intracellular protein content[7,8]. For instance, neutrophils can respond to microorganisms via phagocytosis, generation of reactive oxygen species (ROS), and degranulation or the release of NETs[2]. NETs are defined as extracellular structures composed of DNA and cytosolic, granular and nuclear proteins assembled on a scaffold of decondensed chromatin[1]. NETs contain, neutralize and kill microorganisms including fungi, bacteria and parasites, but they can also be released in response to other stimuli,

including crystals and immune complexes[9–11]. Of note, neutrophils and, in particular, NETs that are released following cell activation are associated with autoimmune pathogenesis, such as in rheumatoid arthritis or systemic lupus erythematosus (SLE)[12,13]. Moreover, NETs have been linked to the development of autoantibodies that are also associated with progression and severity of autoimmune diseases[6,7]. In inflammatory diseases, such as COVID-19 infection, neutrophils can acquire a persistent inflammatory signature that leads to increased NET release, which is associated with disease severity[14–16].

NETs function as immune stimulants, acting as damage-associated molecular patterns that induce local inflammation and tissue damage[7]. NETs contain various molecules recognized by immune cell receptors. For example, following internalization of NETs by macrophages and dendritic cells, the cytosolic sensor cyclic GMP–AMP synthase (cGAS) recognizes the DNA backbone of NETs[17]. In addition, the cell-surface receptor TLR9 also recognizes NET–DNA in a process that is facilitated

[1]MRC Centre for Medical Mycology, University of Exeter, Exeter, UK. [2]Institute of Medical Sciences, University of Aberdeen, Aberdeen, UK. [3]Institute of Infection, Immunity and Inflammation, University of Glasgow, Glasgow, UK. [4]Centre for Arthritis and Musculoskeletal Health, University of Aberdeen, Aberdeen, UK. [5]Joan and Sanford I. Weill Department of Medicine, Weill Cornell Medicine, New York City, NY, USA. [6]The Jill Roberts Institute for Research in Inflammatory Bowel Disease (JRI), Weill Cornell Medicine, New York City, NY, USA. [7]Program in Cell Biology, Hospital for Sick Children, Toronto, Ontario, Canada. [8]School of Cellular and Molecular Medicine, University of Bristol, Bristol, UK. [9]Faculty of Medicine, Department of Microbiology, Infectious Diseases, and Immunology, Laval University, Quebec City, Quebec, Canada. [10]Medical Affairs and Innovation, Canadian Blood Services, Toronto, Ontario, Canada. [11]Centre for Cytomics, University of Exeter, Exeter, UK. [12]Department of Immunology and Inflammation, Imperial College London, London, UK. ✉e-mail: gordon.brown@exeter.ac.uk

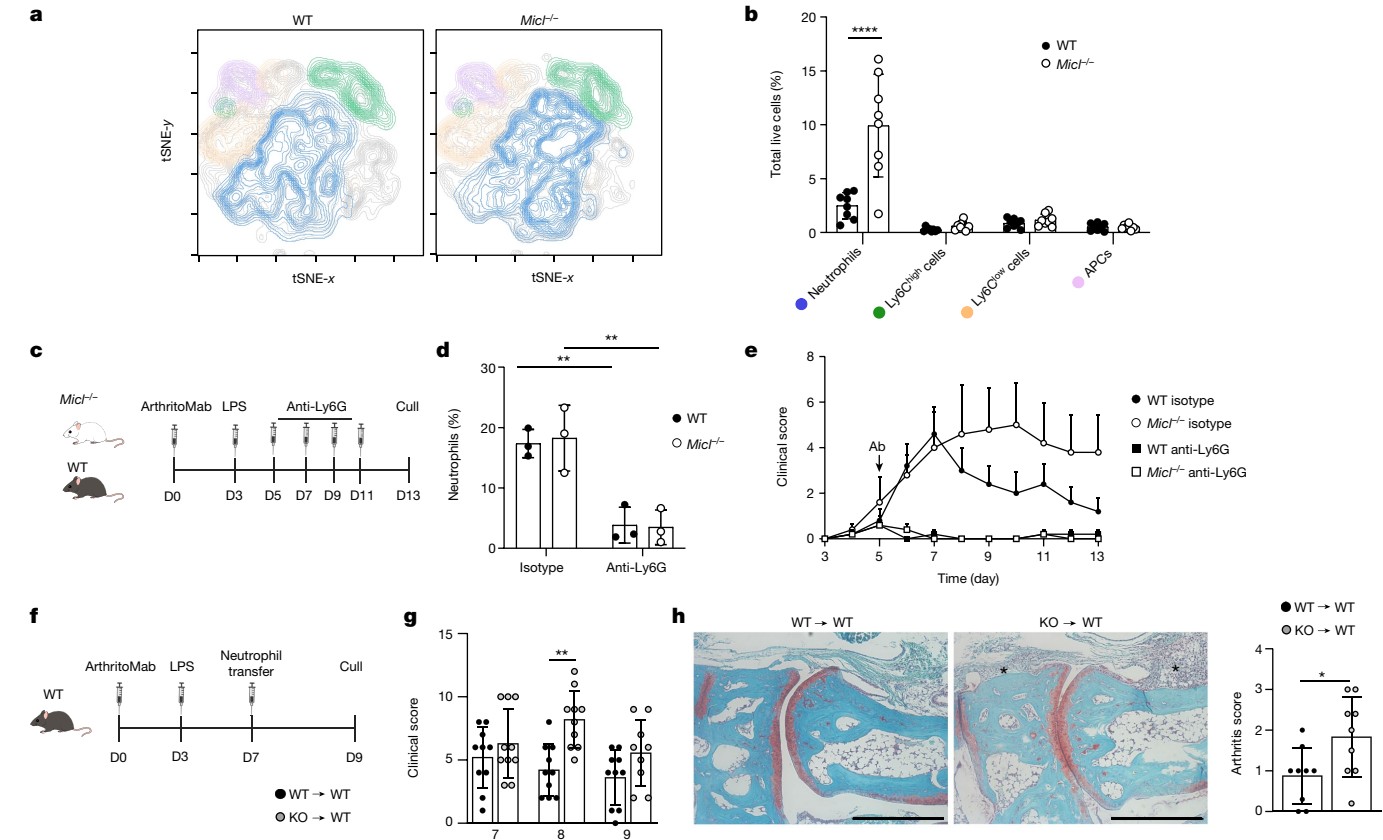

**Fig. 1 | MICL is required for control of neutrophil responses. a**, tSNE plots of CD45[+] myeloid populations in the inflamed ankle joint during CAIA displayed as CD11b[+] cells (grey), neutrophils (blue), Ly6C[high] cells (green), Ly6C[low] cells (orange) and remaining antigen-presenting cells (APCs; pale violet). **b**, Myeloid cell populations (defined as shown in the gating strategy in Extended Data Fig. 1b) in the inflamed ankle joint during CAIA at day 7 are represented as a percentage of total live cells (pooled data from two independent experiments with four mice per group per experiment). WT versus *Micl*[−/−] neutrophils *P* < 0.0001. **c**, Schematic representation of the anti-Ly6G-mediated neutrophil depletion strategy in the CAIA model. D0, day 0; LPS, lipopolysaccharide. **d**, Quantification of neutrophils (CD45[+]CD11b[+]F4/80[−]SSC[high]) in the peripheral blood at day 9 by flow cytometry (*n* = 1 experiment with 3 mice per group). Isotype versus anti-Ly6G WT *P* = 0.0035 and *Micl*[−/−] *P* = 0.0021. **e**, Severity scoring of WT and knockout (*Micl*[−/−]) mice treated with isotype or anti-Ly6G antibodies (Abs), as indicated (*n* = 1 experiment with 5 mice per group). **f**, Schematic

representation of neutrophil adoptive transfer during CAIA in WT mice. KO, knockout. **g**, CAIA severity scoring in WT mice that received adoptively transferred WT or knockout neutrophils, as indicated (pooled data from two independent experiments with five mice per group per experiment). Day 8 WT versus *Micl*[−/−] *P* = 0.0013. **h**, Representative Safranin O-stained sections of the tarsal joints of WT mice that received adoptively transferred WT or knockout neutrophils, as indicated at day 8 (left). Synovial inflammation (black asterisks) is indicated. Scale bars, 500 μm. The histological arthritis severity score is also shown (right; nine mice per group). WT versus *Micl*[−/−] *P* = 0.0281. Data are represented as mean ± s.d. (**b**,**d**,**e**,**g**,**h**). Statistical significance was determined by two-way analysis of variance (ANOVA) with Bonferroni's multiple comparisons test (**b**,**d**,**e**,**g**). Data were analysed using an unpaired two-tailed Student's *t*-test (**h**). *\*P* < 0.05, *\*\*P* < 0.01 and *\*\*\*\*P* < 0.0001. Schematics in panels **c**,**f** were created using BioRender (https://biorender.com).

via two NET proteins: LL-37, which potentiates a type I IFN induction, and HMGB1, which acts through the receptor for advanced glycation products (RAGE)[18–22]. Recently, it has been shown that TLR4 recognizes and is activated by NET-associated histones, whereas histone containing chromatin DNA regulates TLR4 recruitment to endosomes[23]. However, the mechanisms that regulate cellular responses to NETs are still incompletely understood[23].

Loss or mutation of inhibitory receptors are often associated with unchecked inflammation and destructive autoimmunity[24]. Recent work from our group has indicated that MICL (also known as CLEC12A) is one such receptor, which can regulate the pathogenesis of rheumatoid arthritis[25]. In contrast to wild-type (WT) mice, we showed that MICL-deficient animals (*Micl*[−/−]) undergoing collagen antibody-induced arthritis (CAIA) presented with exacerbated joint inflammation that did not resolve[25]. Through investigating the underlying mechanism, we discovered that MICL functions as an essential pattern recognition receptor (PRR) for NETs on neutrophils. Recognition of NETs by MICL inhibits neutrophil activation and further NET formation, regulating a

positive-feedback cycle that, on the one hand, protects from excessive inflammation during multiple autoimmune diseases, but on the other hand, increases susceptibility to invasive infections.

## MICL regulates neutrophil responses

*Micl*[−/−] mice exhibit enhanced and non-resolving joint inflammation compared with WT mice during CAIA[25] (Extended Data Fig. 1a). To understand the underlying mechanism, we first determined the cell type (or types) responsible for this phenotype by analysing the cellular infiltrate in the hind paws of arthritic mice by flow cytometry. Day 7 was selected for analysis, representing the peak clinical score in WT mice undergoing CAIA (Extended Data Fig. 1a). We found a significant increase in neutrophils (defined as CD45[+]Ly6G[+]CD11b[+] cells) in the joints of *Micl*[−/−] animals, whereas other myeloid populations were unaffected (Fig. 1a,b and Extended Data Fig. 1b). Analyses of an earlier time point (day 5) showed that neutrophil numbers were higher in the joints of MICL-deficient mice even during early stages of inflammation, when

there was no apparent clinical difference between the two groups of mice, again without affecting other myeloid populations (Extended Data Fig. 1c). Of note, MICL deficiency also exacerbated joint inflammation in another effector phase model of rheumatoid arthritis: the K/BxN serum transfer model[26] (Extended Data Fig. 2a). As in CAIA, $Micl^{-/-}$ mice in the K/BxN serum transfer model presented higher numbers of neutrophils than WT mice (Extended Data Fig. 2b). There were no alterations in the frequency of cellular populations in the bone marrow or blood of naive $Micl^{-/-}$ mice (Extended Data Fig. 2c,d).

We next analysed the expression of key neutrophil adhesion and activation molecules (CD18, CD11b and CD62L) and chemotactic receptors (CXCR2 (ref. 27), CCR1 and C5aR[28,29]) by flow cytometry. Neutrophils in the joints of $Micl^{-/-}$ mice under CAIA exhibited increased expression of CD18 and CD11b, and decreased expression of the lymphoid organ homing receptor CD62L, compared with WT animals, indicative of a heightened state of activation (Extended Data Fig. 2e). This activated phenotype was only observed in cells isolated from the joints and not on neutrophils concomitantly isolated from the peripheral blood of mice undergoing CAIA (Extended Data Fig. 2e). Analysis of receptors controlling neutrophil migration revealed increased expression of C5aR and the chemokine receptor CXCR2 on neutrophils isolated from the joints of $Micl^{-/-}$ mice, compared with WT animals, although the increase of the former was not statistically significant (Extended Data Fig. 2f). By contrast, expression levels of CXCR2 and C5aR were comparable between neutrophils isolated from the peripheral blood of WT and $Micl^{-/-}$ mice (Extended Data Fig. 2f). The expression of CCR1 on neutrophils was unaffected by MICL deficiency (Extended Data Fig. 2f). MICL-deficient animals in the K/BxN serum transfer model of arthritis showed similar changes in neutrophil activation (Extended Data Fig. 2g). There were no differences in the expression of key adhesion and activation molecules and chemotactic receptors in the bone marrow or blood neutrophils of naive animals (Extended Data Fig. 2h,j). Together, these data demonstrate that MICL deficiency is characterized by an increased number and an enhanced activation of neutrophils in the joints of mice during CAIA and in the K/BxN serum transfer model of arthritis.

Neutrophils contribute significantly to the pathology of CAIA[30]. To investigate whether these cells were solely responsible for the exacerbated disease in $Micl^{-/-}$ mice, we successfully depleted circulating neutrophils via anti-Ly6G administration following the onset of CAIA (Fig. 1c,d and Extended Data Fig. 3a). Of note, clinical disease was significantly reduced following neutrophil depletion, and to an equivalent level, in both WT and $Micl^{-/-}$ mice (Fig. 2e). A similar reduction in disease severity occurred in both groups of mice following neutrophil depletion using the less-specific marker GR-1 (Extended Data Fig. 3b–e). These data show that the elevated pathology in the $Micl^{-/-}$ mice undergoing CAIA was stemming from the neutrophil compartment.

We then investigated whether the aberrant neutrophil response in the $Micl^{-/-}$ mice resulted from a cell-intrinsic or cell-extrinsic defect. For this, we isolated neutrophils from the bone marrow of naive WT or $Micl^{-/-}$ mice (Extended Data Fig. 3f) and performed a single adoptive transfer of these cells into arthritic WT mice at day 7 post-induction of CAIA (Fig. 1f). We detected transferred cells in the joints even 48 h post-transfer (Extended Data Fig. 3g). Although transfer of WT neutrophils did not significantly alter disease severity (Fig. 1g,h), transfer of MICL-deficient neutrophils into WT mice induced a significant increase in joint inflammation, determined by both clinical score and histological changes (Fig. 1g,h). Together, these data reveal that aberrant neutrophil function underlies the pathology of $Micl^{-/-}$ mice during CAIA.

## MICL controls NET formation

We then determined how loss of MICL was affecting neutrophil function. A previous report has suggested that MICL negatively regulates the respiratory burst in neutrophils in response to specific ligands,

such as uric acid crystals (monosodium urate (MSU))[31]. Indeed, we recapitulated this observation and showed that $Micl^{-/-}$ neutrophils induced a significantly elevated ROS in response to MSU, but not to phorbol-12-myristate-13-acetate (PMA) or zymosan, a potent inducer of ROS and a ligand for a related C-type lectin (Dectin-1)[32], respectively (Fig. 2a). $Micl^{-/-}$ neutrophils also produced higher levels of ROS in response to *A. fumigatus* hyphae, which are also known to induce ROS[33] (Fig. 2a). As ROS production is involved in induction of NETs[34], we examined this programmed cell death response and found that neutrophils from $Micl^{-/-}$ mice exhibited significantly increased NET formation in vitro following MSU stimulation (Fig. 2b and Extended Data Fig. 4,b). NET formation was unaltered following PMA stimulation (Fig. 2b and Extended Data Fig. 4a). The ability of diphenyleneiodonium chloride (DPI) to inhibit NET formation in cells from MICL-deficient animals revealed the dependence of this response on NADPH oxidase (Extended Data Fig. 4c). MICL transduces intracellular inhibitory signals through SHP1/2 (refs. 35,36); we found that inhibition of SHP1/2 increased NET formation in WT neutrophils, but not in MICL-deficient neutrophils, indicating that inhibitory signalling from MICL was required to regulate this response (Extended Data Fig. 4d). Of note, we detected increased NET formation (defined as GR-1+cit-H3+DNA/H1+-stained cells) in the joints of $Micl^{-/-}$ mice undergoing CAIA at day 11 by immunofluorescence (Fig. 2c and quantification in Extended Data Fig. 4e) and by imaging flow cytometry (Fig. 2e and Extended Data Fig. 4f).

In addition to ROS, the formation of NETs can require histone citrullination through protein arginine deaminase 4 (PAD4)[37]. Using two different PAD4 inhibitors, BB-CL-amidine[38] and GSK484 (ref. 39), complete inhibition of NET formation in vitro was shown (Extended Data Fig. 5a). In vivo (Fig. 2d), imaging flow cytometry revealed that treatment with these inhibitors led to a significant reduction in the number of NETs detectable in the joints of $Micl^{-/-}$ but not WT mice during CAIA (Fig. 2e). There was no effect of the inhibitor on the number of neutrophils recruited into the joints of these mice (Fig. 2f). Treatment of $Micl^{-/-}$ mice with both PAD4 inhibitors reduced the severity of the disease back to WT levels (Fig. 2g and Extended Data Fig. 5b). There was no significant effect of the inhibitors on disease development in WT mice (Fig. 2g and Extended Data Fig. 5b), as expected[40]. Specific PAD4 inhibition also reduced disease severity in MICL-deficient mice during the K/BxN serum transfer model of arthritis (Extended Data Fig. 5c). Administration of DNase I to degrade NETs in vivo ameliorated the enhanced clinical severity in the $Micl^{-/-}$ mice undergoing CAIA, but had no effect on disease severity in WT mice (Extended Data Fig. 5d). Thus, these data show that aberrant regulation of NET formation underlies the pathology of MICL-deficient mice during models of arthritis.

## Anti-MICL antibodies link to NET diseases

We had previously shown that administration of antibodies targeting MICL was able to exacerbate CAIA in WT mice, recapitulating the phenotype of MICL-deficient animals[25]. Here we found that treatment of WT neutrophils with anti-MICL antibodies resulted in elevated ROS production in response to MSU, but not zymosan, suggesting that these antibodies were blocking receptor functionality (Extended Data Fig. 6a). To gain more insight into the effect of anti-MICL antibodies on neutrophil activation, we administered anti-MICL monoclonal antibodies to WT mice during CAIA in the presence of NET inhibitors. As previously observed, administration of anti-MICL antibodies exacerbated CAIA pathology in WT mice (Extended Data Fig. 6b). Administration of a PAD4 inhibitor completely reduced disease symptoms to the levels seen in the isotype control-treated mice (Extended Data Fig. 6b), revealing that aberrant NET formation was underlying the effect of the anti-MICL antibodies on disease pathology.

To further substantiate the receptor-blocking effect of anti-MICL antibodies, we tested them using an alternative model: collagen-induced arthritis (CIA). This is an important T cell-dependent model

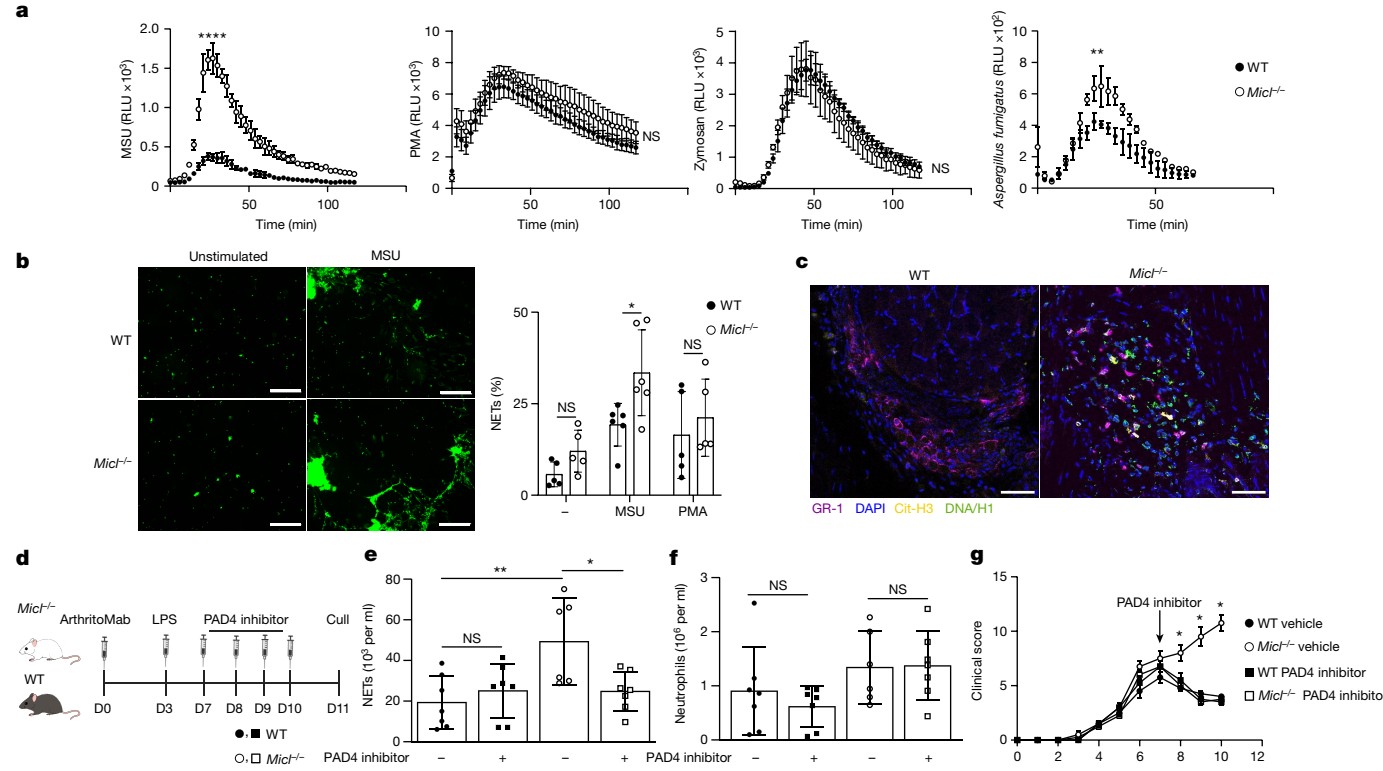

**Fig. 2 | NET formation drives CAIA severity in *Micl*⁻/⁻ mice. a**, ROS generation by MSU, PMA, zymosan or *A. fumigatus* hyphae-stimulated bone marrow neutrophils, depicted as relative light units (RLU), over time. Data are a representative example of *n* = 4 independent experiments and mean ± s.d. performed in triplicate. Area under curve was analysed using unpaired two-tailed Student's *t*-test. WT versus *Micl*⁻/⁻ MSU *P* < 0.0001, *A. fumigatus P* = 0.0056 and not significant (NS). **b**, Sytox green fluorescent images of MSU-induced NETs and NET formation percentage by thioglycollate-elicited neutrophils (3 fields of view per condition) 4 h post-stimulation with PMA or MSU, or medium alone (−). Data are represented as mean ± s.d. (*n* = 3 independent experiments performed in triplicate). Scale bars, 500 μm. Statistical significance was determined by two-way ANOVA with Bonferroni's multiple comparisons test. WT versus *Micl*⁻/⁻ MSU *P* = 0.0304. **c**, Representative confocal immunofluorescence microscopy of NETs in CAIA WT and *Micl*⁻/⁻ day 11 synovial sections (as per Extended Data Fig. 1a; *n* = 1 experiment with 3 mice per group). GR-1 (purple), DAPI (blue), citrullinated histone 3 (cit-H3; yellow) and DNA/H1 (green) are shown. NETs are defined as GR-1⁺cit-H3⁺DNA/H1⁺-stained cells. Scale bars, 100 μm. **d**, Schematic of the PAD4 inhibitor CAIA treatment regime. **e,f**, Image stream quantification of NET-positive cells (**e**) and neutrophils isolated from arthritic ankle joints (**f**) at day 11 during PAD4 inhibitor (BB-CL-amidine) treatment. Pooled data are from 2 independent experiments with *n* = 7 biologically independent mice per group, represented as mean ± s.d. Statistical significance was determined using two-way ANOVA with Tukey's multiple comparisons test. WT versus *Micl*⁻/⁻ *P* = 0.0063 and *Micl*⁻/⁻ versus *Micl*⁻/⁻ + PAD4 inhibitor *P* = 0.0295 (**e**). **g**, Severity scoring of WT and *Micl*⁻/⁻ mice during CAIA treated with vehicle or the PAD4 inhibitor (GSK484). The black arrow indicates the start of treatment. Data shown are a representative example of *n* = 2 experiments with 4 mice per group, represented as mean ± s.d. Statistical significance was determined by two-way ANOVA with Tukey's multiple comparisons test. Days 8–10 *Micl*⁻/⁻ versus *Micl*⁻/⁻ + GSK484 *P* < 0.0001. *P* < 0.05, **P* < 0.01 and ****P* < 0.0001. The diagram in panel **d** was created using BioRender (https://biorender.com).

of arthritis that recapitulates both the pathophysiology and the histological presentation of human disease, although C57BL/6 mice, the same background on which our knockout model was based, are resistant to this model of disease[41]. However, the ability of anti-MICL antibodies to alter MICL function enabled us to explore the role of this receptor during CIA induced in DBA/1 mice. As before, administration of anti-MICL antibodies to DBA/1 mice exacerbated CIA disease and increased the number of neutrophils in the joints (Extended Data Fig. 6c). These results confirm that antibodies to MICL interfere with key functions of this receptor and its ability to regulate neutrophil activation, exacerbating arthritic inflammation in different mouse models.

As we previously detected the presence of anti-MICL antibodies in a small cohort of patients with rheumatoid arthritis[25], we explored whether these antibodies could affect neutrophil activation. Here we found that anti-human MICL antibodies were able to enhance the respiratory burst and NET formation in human neutrophils in vitro in response to MSU, but not to PMA (Fig. 3a,b). Moreover, serum samples from patients with rheumatoid arthritis were able to increase ROS

production in human neutrophils following stimulation with MSU (Fig. 3c and Extended Data Fig. 6d); this effect directly correlated with the anti-MICL serum titres, demonstrating that patient anti-MICL antibodies interfere with MICL function.

To determine whether anti-MICL antibodies were influencing clinical severity, we further analysed the prevalence of these antibodies in patients with rheumatoid arthritis, using an ELISA-based assay[25]. For this, we interrogated 200 serum samples from patients, along with matched controls, obtained from the Scottish Early Rheumatoid Arthritis (SERA) cohort[42]. We found that the majority of patients with rheumatoid arthritis presented with significantly elevated titres of anti-MICL autoantibodies (Fig. 3d). High levels of autoantibodies were also detected in some healthy controls without rheumatoid arthritis. Although we did not find an association between the level of anti-MICL autoantibodies and DAS28 or Sharp erosion scores (data not shown), after correcting for age and sex, we detected an association between the level of anti-MICL antibodies and the level of rheumatoid factor, an autoantibody targeting the Fc region of IgG antibodies, which is used to diagnose, classify and predict development of rheumatoid

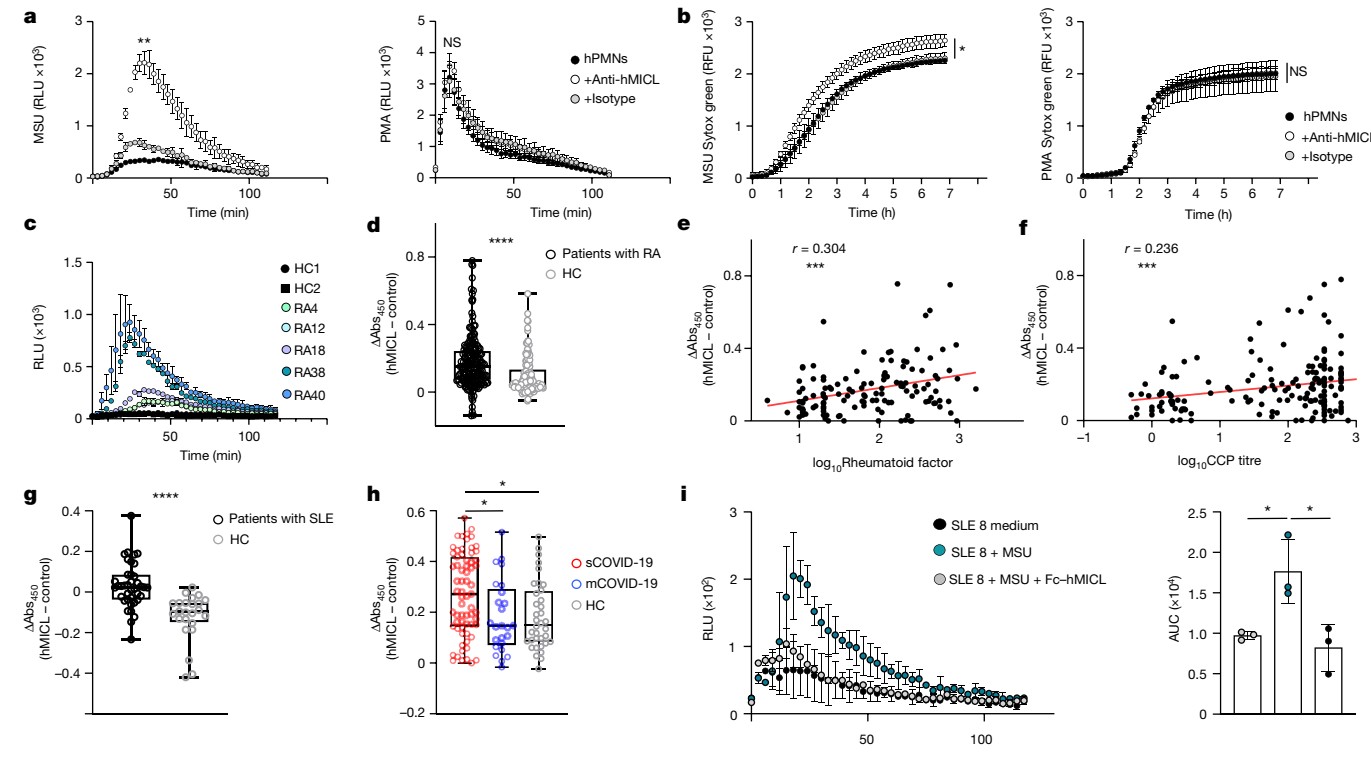

**Fig. 3 | Anti-MICL antibodies modulate neutrophil function and correlate with disease. a**, ROS generation by MSU or PMA of human neutrophils (hPMNs) in the presence or absence of anti-MICL antibodies (anti-hMICL), depicted as RLU over time. Anti-hMICL versus isotype MSU $P = 0.0084$. **b**, Fluorescence of NET-bound Sytox green of MSU or PMA-induced NETs in hPMNS in the presence or absence of anti-hMICL. MSU anti-hMICL versus isotype $P = 0.0043$. RFU, relative fluorescence units. **c**, ROS generation by MSU of hPMNs in the presence of serum from patients with rheumatoid arthritis (RA) and healthy controls (HC). HC1 versus RA38 $P < 0.0001$, HC1 versus RA40 $P < 0.0001$, HC2 versus RA38 $P < 0.0001$ and HC2 versus RA40 $P < 0.0001$. **d**, Level of anti-MICL autoantibodies detected in serum samples from patients with rheumatoid arthritis ($n = 199$) and healthy controls ($n = 132$). $Abs_{450}$, absorbance at 450 nm. **e**, Correlation of MICL autoantibody titres with rheumatoid factor levels in SERA cohort serum from patients with rheumatoid arthritis. **f**, Correlation of MICL autoantibody titres with CCP levels in SERA cohort serum from patients with rheumatoid arthritis. **g**, Level of anti-MICL autoantibodies detected in

serum from patients with SLE ($n = 40$) and healthy controls ($n = 25$). **h**, Level of anti-MICL autoantibodies detected in serum from patients with mild/moderate (mCOVID-19; $n = 25$) and severe (sCOVID-19; $n = 67$) COVID-19 and healthy controls ($n = 36$). sCOVID-19 versus mCOVID-19 $P = 0.0472$ and sCOVID-19 versus healthy controls $P = 0.0284$. **i**, ROS generation and area under curve (AUC) of hPMNs stimulated with MSU in the presence of serum from patients with SLE pre-incubated with Fc–hMICL. Data are a representative example of $n = 3$ (**a,b**) or $n = 2$ (**c,i**) independent experiments, and mean ± s.d. performed in triplicate. The AUC was analysed by one-way ANOVA with Tukey's multiple comparisons test. Box plots extend from the 25th to 75th percentile, including the median, and whiskers extend from the minimum to maximum value (**d,g,h**). Data were analysed by unpaired two-tailed Student's $t$-test (**d,g**), a Spearman correlation and two-sided $t$-test (**e,f**), or using one-way ANOVA with Kruskal–Wallis multiple comparisons test (**h**). *$P < 0.05$, **$P < 0.01$, ***$P < 0.001$ and ****$P < 0.0001$.

arthritis[43,44] (Fig. 3e). We also found a significant correlation between the level of anti-MICL autoantibodies and the level of anti-cyclic citrullinated peptide (CCP) antibodies in patients with rheumatoid arthritis (Fig. 3f), indicative of a direct link of serum anti-MICL antibodies with NETosis and disease severity[45,46]. Together, these results show that autoantibodies targeting MICL interfere with the function of this receptor and correlate with severity of rheumatoid arthritis in patients.

We next determined whether anti-MICL autoantibodies were associated with any other NET-linked inflammatory diseases. Of note, we found high titres of anti-MICL antibodies in patients with SLE (Fig. 3g) and severe COVID-19 (Fig. 3h), both of which are inflammatory disorders where disease severity has been linked to NETosis[15,47]. Moreover, as we found in rheumatoid arthritis, serum samples containing anti-MICL antibodies from patients with SLE or severe COVID-19 were able to modulate neutrophil function, as demonstrated by increased ROS production following MSU stimulation (Extended Data Fig. 6e,f). We demonstrated the specificity of this response through the addition of a soluble chimeric protein containing the C-type lectin-like domain (CTLD) of human MICL, which abrogated the serum effect

on the neutrophil response (Fig. 3i and Extended Data Fig. 6g). A related control CTLD had no effect on these responses (Extended Data Fig. 6g). Thus, autoantibodies that modulate MICL-mediated neutrophil functions are present in patients with a wide variety of NET-related pathologies.

## MICL is a PRR for NETs

Given that NETs can activate pro-inflammatory functions of naive neutrophils[48], we wondered whether antibody-mediated interference or loss of MICL function was affecting neutrophil responses to NETs themselves. We observed that following exposure to preformed NETs, MICL-deficient neutrophils induced significantly increased levels of ROS and higher formation of NETs compared with the response of WT neutrophils (Fig. 4a,b and Extended Data Fig. 7a,b). ROS production in MICL-deficient cells was unaltered when neutrophils were stimulated with NETs treated with polymyxin B (Extended Data Fig. 7c). Following stimulation of cells from both groups of mice with preformed NETs, the ability of both ROS (DPI) and NET (PAD4) inhibitors to prevent NET formation shows that this process is dependent on NADPH and

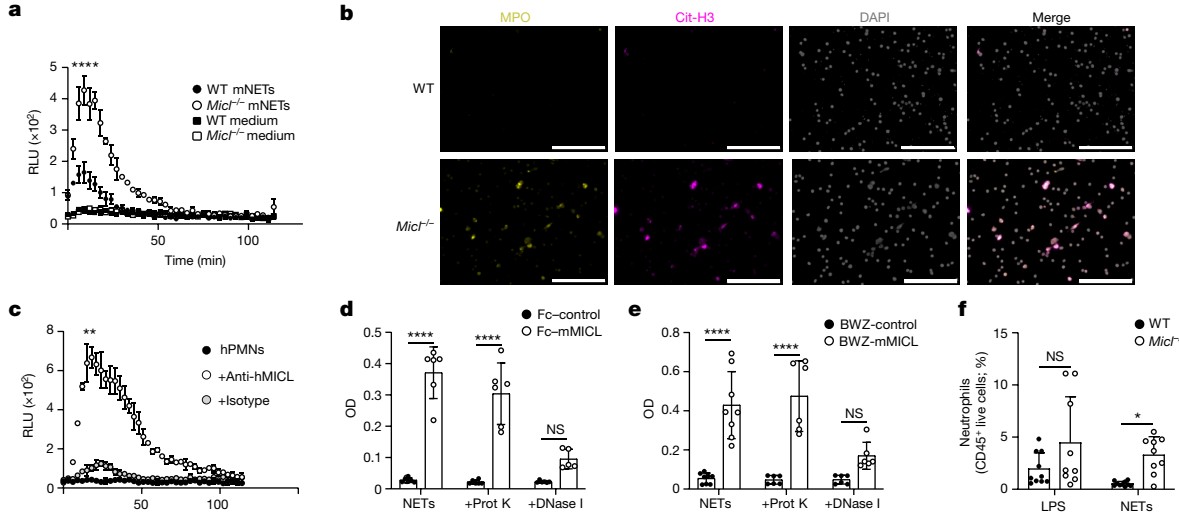

**Fig. 4 | NET–MICL interaction regulates inflammation. a**, ROS generation by bone marrow neutrophils stimulated with preformed NETs, depicted as RLU over time. Data are a representative example of $n = 4$ independent experiments, depicted as mean ± s.d. performed in triplicate. mNET, preformed mouse NETs. **b**, Immunofluorescence staining for MPO (yellow), cit-H3 (magenta) and DNA (DAPI; grey) of neutrophils stimulated with preformed NETs. Scale bars, 200 µm. Quantification is in Extended Data Fig. 7b. **c**, ROS generation by human neutrophils stimulated with preformed NETs in the presence or absence of antibodies targeting MICL. Data are a representative example of $n = 2$ independent experiments, depicted as mean ± s.d. performed in duplicate. hPMNs versus anti-hMICL $P = 0.0011$ and anti-hMICL versus isotype $P = 0.0016$. **d**, Fc–MICL recognition of untreated NETs (NETs), proteinase K-treated (+prot K) or DNase-treated (+DNase I) NETs by ELISA. **e**, MICL-expressing BWZ reporter cell recognition of untreated NETs, proteinase K-treated or DNase-treated NETs. OD, optical density. Pooled data from two independent experiments, depicted as mean ± s.d. performed in triplicate (**d**,**e**). **f**, Neutrophil infiltration (CD45⁺CD11b⁺Ly6G⁺ cells) 4 h after preformed NETs or LPS injection in the peritoneum of WT and *Micl⁻/⁻* mice. Pooled data are from two independent experiments ($n = 9$ mice per group), depicted as mean ± s.d. NETs WT versus *Micl⁻/⁻* $P = 0.0384$. *$P < 0.05$, **$P < 0.01$ and ****$P < 0.0001$.

PAD4 activity (Extended Data Fig. 7d). To show that human MICL was functioning similarly, we generated MICL-knockout human neutrophils derived from CD34⁺ haematopoietic progenitors[49] (Extended Data Fig. 7e) and found increased ROS production in response to preformed NETs compared with control cells (Extended Data Fig. 7f). Of note, we also demonstrated that normal neutrophils isolated from healthy human controls had significantly elevated ROS responses to human NETs in the presence of anti-human MICL antibodies (Fig. 4c).

To determine whether MICL was functioning as a PRR for NETs themselves, we examined the ability of a soluble chimeric protein consisting of the CTLD of mouse MICL fused to the Fc region of human IgG1 (Fc–mMICL) to directly recognize NETs. Using this Fc–protein and a structurally related control protein, we showed that MICL binds directly to NETs using an ELISA-based assay (Fig. 4d). Moreover, recognition of NETs by MICL could also be demonstrated in a cellular context using MICL-expressing reporter cells[50] (Fig. 4e). MICL recognition of NETs was blocked in the presence of antibodies targeting MICL (Extended Data Fig. 7g), showing that antibodies to MICL block the ability of the receptor to recognize its ligand. To determine which component of NETs is required to interact with this receptor, we treated NETs with DNase I or proteinase K, and found that treatment with DNase abolished the ability of MICL to recognize NETs (Fig. 4d,e). Furthermore, MICL-deficient neutrophils stimulated with DNase-treated NETs induced significantly lower levels of ROS (Extended Data Fig. 7h). Consistent with these observations, MICL was able to recognize genomic DNA, and this interaction could be blocked in the presence of anti-MICL antibodies (Extended Data Fig. 7i).

To show a direct role for MICL in response to NETs in vivo, we injected preformed NETs into the peritoneum of *Micl⁻/⁻* and WT mice. Neutrophil recruitment was significantly higher in *Micl⁻/⁻* mice treated with NETs, but similar to WT in mice administered with LPS (Fig. 4f), revealing a specific negative regulatory function of MICL following recognition of NETs. Moreover, there was a trend to higher levels of cell-free DNA measurements following NET administration in MICL-deficient mice (Extended Data Fig. 7j). Collectively, our data show that MICL functions as a PRR for NETs, regulating activation and NET formation in both mouse and human neutrophils.

## MICL regulates fungal-induced NETs

To validate our hypothesis that NET sensing by MICL functions as a universal regulator of neutrophil activation and NET formation, we examined neutrophil responses elicited by *A. fumigatus*. NET formation is critical in the immune response to *A. fumigatus*, restraining fungal growth and preventing tissue dissemination[51,52]. We demonstrated that NETs are released in response to *A. fumigatus* hyphae[53,54] (Fig. 5a), and both MICL-deficient neutrophils and human neutrophils treated with anti-MICL antibodies induce higher levels of NET formation in vitro in response to this pathogen (Fig. 5a and Extended Data Fig. 8a). Of note, MICL-deficient mice were significantly more protected than WT mice following intravenous infection with *A. fumigatus* conidia (Fig. 5b). This increased susceptibility of WT mice was associated with increased fungal burdens in the brains of these animals at day 2 post-infection (Fig. 5c and Extended Data Fig. 8b). Analysis of inflammatory responses showed that serum levels of IL-6 and brain levels of G-CSF were significantly higher in infected WT than in MICL-deficient mice (Fig. 5d,e), showing that MICL functions to restrain systemic inflammation during invasive aspergillosis. Furthermore, we confirmed that the increased survival of MICL-deficient mice was associated with higher NET formation, as the treatment of these animals with a PAD4 inhibitor (GSK484), but not with the vehicle, increased the susceptibility of the knockout mice to *A. fumigatus* infection to levels similar to those observed in WT mice (Fig. 5f and vehicle control shown in Extended Data Fig. 8c). Moreover, administration of the inhibitor increased fungal burdens in the brains of MICL-deficient mice (Fig. 5g). There was no effect of the

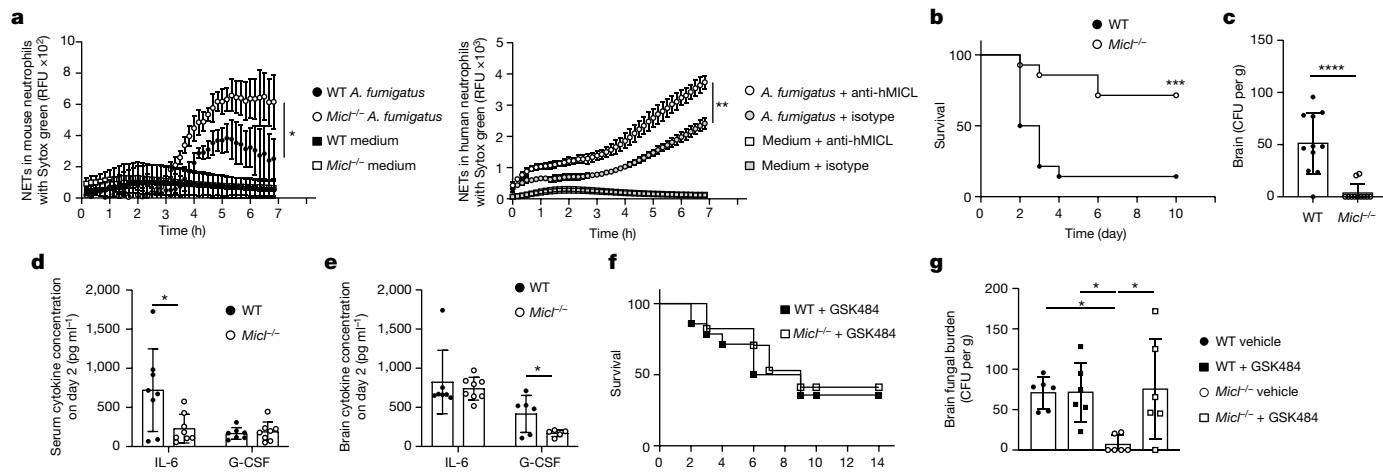

**Fig. 5 | MICL regulates NET formation during fungal infection. a**, Fluorescence of *A. fumigatus* hyphae-induced NETs bound to Sytox green in bone marrow-isolated neutrophils from WT and MICL-deficient mice or human neutrophils in the presence of antibodies to MICL. The fluorescence background signal from the unstimulated controls was subtracted from values. Data are a representative example of *n* = 3 independent experiments, depicted as mean ± s.d. performed in triplicate. The AUC was analysed using an unpaired two-tailed Student's *t*-test. WT versus *Micl*[−/−] *P* = 0.0272 and anti-hMICL versus isotype *P* = 0.0032. **b**, Survival of mice following intravenous infection with 10[6] *A. fumigatus* conidia (*n* = 15 mice per group). Pooled data are from two independent experiments, analysed by log-rank test; *P* = 0.0005. **c**–**e**, Brain fungal burdens (**c**), and serum (**d**) and brain (**e**) cytokine levels of mice 2 days after intravenous infection with 10[6] *A. fumigatus* conidia. Values are mean ± s.e.m. of pooled data from two independent

experiments. *n* = 12 (**c**) or *n* = 8 (**d**,**e**) biologically independent mice, analysed using an unpaired two-tailed Student's *t*-test. IL-6 WT versus *Micl*[−/−] *P* = 0.0185 (**d**) and G-CSF WT versus *Micl*[−/−] *P* = 0.0492 (**e**). CFU, colony-forming unit. **f**, Survival of mice following intravenous infection with 10[6] *A. fumigatus* conidia and treated with GSK484 (PAD4 inhibitor). Pooled data are from two independent experiments; *n* = 14 biologically independent mice, analysed by log-rank test. **g**, Brain fungal burdens of mice 2 days after intravenous infection with 10[6] *A. fumigatus* conidia treated with GSK484 (PAD4 inhibitor) or vehicle control. Values are mean ± s.e.m. of 1 experiment with *n* = 6 biologically independent mice, analysed by one-way ANOVA with Tukey's multiple comparisons test. WT vehicle versus *Micl*[−/−] vehicle *P* = 0.0378, WT + GSK4848 versus *Micl*[−/−] vehicle *P* = 0.0356 and *Micl*[−/−] vehicle versus *Micl*[−/−] + GSK484 *P* = 0.0242. **P* < 0.05, ***P* < 0.01, ****P* < 0.001 and *****P* < 0.0001.

NET inhibitor or vehicle control on disease development and fungal burdens in WT mice[53] (Fig. 5g and Extended Data Fig. 8c). Thus, dysregulation of NET formation in MICL-deficient mice results in increased resistance to a systemic fungal infection.

## Discussion

The integration of signalling between diverse receptors helps to refine and modulate immune responses[55]. Inhibitory receptors regulate immune cell activation, avoiding excessive inflammation and host tissue damage. Our data revealed that MICL, an inhibitory C-type lectin receptor, is vital for the regulation of key neutrophil functions. We discovered that MICL is a PRR for NETs, and that this interaction is essential for controlling key neutrophil responses, including the respiratory burst and further induction of NETs themselves. This regulation is critical for limiting NET-mediated inflammatory conditions, such as in arthritis, SLE or severe COVID-19 infections, in which MICL prevents the formation of a positive-feedback loop that leads to uncontrolled inflammation (Extended Data Fig. 9). By contrast, this regulation of NET formation supresses robust control of invasive infections, promoting disease susceptibility.

In the context of arthritis, we showed that loss of MICL functionality increases NET formation and neutrophil activation by NETs, triggering an inflammatory chain reaction that leads to increased joint inflammation in three mouse models of arthritis. We also found that anti-MICL antibodies block the receptor function, and that the presence of these antibodies influences the severity of disease in mouse models and in patients with rheumatoid arthritis. In fact, we demonstrated that serum samples from patients with rheumatoid arthritis possessing anti-MICL autoantibodies, and antibodies to mouse or human MICL, are able to recapitulate MICL-deficient neutrophils, in terms of dysregulation of ROS production and NET release. NETs are released locally in the inflamed joints of patients with rheumatoid

arthritis, and the citrullinated neo-epitopes generated during this process promote the formation of anti-CCP antibody that contribute to the perpetuation of the disease[6,56]. Of note, we found that the level of autoantibodies to MICL correlate with the level of anti-CCP antibodies in patients, indicating a direct link between MICL function and NET formation in patients with rheumatoid arthritis. Moreover, we showed that autoantibodies capable of blocking MICL function are also present in other autoinflammatory conditions, including SLE and severe COVID-19, in which NETs are linked to disease pathology[15,47]. Thus, MICL represents a universal novel autoregulatory pathway that prevents aberrant neutrophil activation and subsequent tissue damage in autoimmunity.

Neutrophils also have a crucial role in protection against fungal pathogens, such as *A. fumigatus*[57]. Protective neutrophil responses to *A. fumigatus* hyphae involve ROS production and NET formation[51,52,54]. Although PAD4 is required for NET formation in response to *A. fumigatus*, previous studies using *Pad4*-knockout mice have suggested no role for this enzyme during infection[53]. Indeed, we found that PAD4 inhibition does not affect disease severity in WT mice. Here we discovered that the PAD4 pathway is stringently regulated by MICL, but that mice lacking this receptor use this pathway to restrict fungal infection through increased NET formation. Pharmacological inhibition of NET formation using a PAD4 inhibitor blocked this pathway in MICL-deficient mice, reverting their susceptibility to infection back to WT levels. Our data show that similar MICL-mediated control of the PAD4 pathway also occurs during autoimmune disease. Indeed, the PAD4 pathway has previously been shown not to be required for effector phase responses during arthritis[58], as we found in our WT mice (Fig. 2g and Extended Data Fig. 5b). Of note, our data show that in the absence of MICL (or inhibition of receptor function by anti-MICL antibodies), the PAD4 pathway becomes activated during autoimmune disease, leading to dysregulated NET formation and aberrant inflammation.

In conclusion, we defined a key mechanism by which MICL regulates neutrophil activation and NET formation. MICL directly recognizes NETs, inducing intracellular signalling that dampens neutrophil responses and preventing the formation of a positive-feedback loop leading to further NET formation. In patients with rheumatoid arthritis, SLE and severe COVID-19, this feedback loop becomes dysregulated by autoantibodies to MICL, leading to a worsened disease outcome. It is likely that autoantibodies to MICL are also associated with the severity of other NET-mediated autoimmune disorders, such as anti-neutrophil cytoplasmic antibody-associated vasculitis. Given that we also observed elevated levels of anti-MICL antibodies in a proportion of healthy individuals, future work should explore the possibility that the presence of such autoantibodies are associated with, or predispose towards, the development of autoimmune disease. Our data also reveal that NET sensing by MICL can have different clinical outcomes depending on the context. In this sense, our results suggest therapeutic targeting of MICL with antibodies to increase ROS and NET formation, contributing to fungal clearance, could be used to treat disseminated forms of invasive infection. Conversely, blocking of the antibody–receptor interaction could be of therapeutic benefit for NET-mediated inflammatory disease. In summary, our observations revealed a novel and fundamental function of inhibitory pathways underlying infectious and non-infectious disease pathogenesis.

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

# Methods

## Mice

*Micl*[−/−] and C57BL/6J mice were bred and maintained under specific pathogen-free (SPF) conditions at the University of Aberdeen, University of Exeter and Charles River Laboratories. Mice were housed in separate groups with bedding exchanges between cages every 2 days for 1 week before commencement of experiments, and maintained on a 12 h–12 h dark–light cycle (07:00–19:00) at 20–24 °C and relative humidity of 55 ± 15%. Mouse experiments were performed by random assignation of age-matched (6–8 weeks old) and sex-matched mice in experimental or control groups at the beginning of each experiment; females were co-housed and experiments were not blinded. All experiments conformed to the ethical review committee of the University of Aberdeen, University of Exeter and the UK Home Office regulations (project license numbers: P79B6F297 and P6A6F95B5).

## CAIA

Male mice received intraperitoneal (i.p.) injections of 2 mg of ArthritoMab monoclonal antibody cocktail (MD Bioproducts) on day 0, followed by 50 μg of lipopolysaccharide (LPS) i.p. (MD Biosciences) on day 3. Joint inflammation was scored visually using a scale of 0 (no visible signs of redness or swelling) to 4 (extensive swelling with signs of deformity). To achieve neutrophil depletion during CAIA, 500 μg of rat anti-mouse Ly6G (clone 1A8, Bio-X-Cell) or isotype control (rat anti-mouse IgG2a; Bio-X-Cell) was administered i.p. to mice every 48 h from day 5 onwards. Alternatively, 50 μg of rat anti-mouse GR-1 (clone RB6-8C5) or isotype control (rat anti-mouse IgG2b) was administered i.p. every 48 h from day 5 onwards. Mice were culled on day 5, 7, 10, 11 or 13, as indicated in the text. For adoptive transfer experiments, bone marrow neutrophils were isolated as detailed below and stained with cell proliferation dye eFluor 670 (eBioscience). Labelled cells ($5 \times 10^6$) were transferred intravenously (i.v.) to WT mice on day 7 following induction of CAIA. Mice were culled on day 9. To investigate the role of NET formation, mice undergoing CAIA were injected i.p. with 2 mg kg$^{-1}$ daily of BB-CL-amidine (Cayman Chemical), 4 mg kg$^{-1}$ daily of GSK484 (Cambridge Biosciences), 75 U per animal of DNase I (Merck) or vehicle (5% DMSO in 10% cyclodextran for BB-Cl-amidine or ethanol 99.9% diluted 1:50 in 0.9% NaCl for GSK484) from day 7 to day 10. Mice were culled on day 11.

## K/BxN serum transfer model of arthritis

Male mice were administered i.p. with 100 μl of serum from transgenic K/BxN mice[26]. The development of clinical symptoms were monitored daily and mice were culled on day 10. Joint inflammation was scored visually using a scale of 0 (no visible signs of redness or swelling) to 4 (extensive swelling with signs of deformity). To investigate the role of NET formation, mice were injected i.p. with 4 mg kg$^{-1}$ daily of GSK484 or vehicle (ethanol 99.9% diluted 1:50 in 0.9 % NaCl) from day 7 to day 10. Mice were culled on day 11.

## CIA

Male DBA/1 mice were purchased from Inotiv and maintained at the University of Exeter. Mice were treated subcutaneously with 100 μl of Immunization Grade Chick Type II Collagen (Chondrex; 200 μg per mouse) in Complete Freund's Adjuvant (Sigma-Aldrich) at day 0 and with 100 μl of Immunization Grade Chick Type II Collagen (200 μg per mouse) in Incomplete Freund's Adjuvant (Sigma-Aldrich) at day 14, followed by an i.p. injection of 50 μg of LPS at day 26. Of anti-MICL antibodies, 0.7 mg of isotype control antibodies or PBS were administered i.p. every 48 h from day 17 to day 33. Mice were culled at day 35. In our animal facility, naive DBA/1 animals did not develop any spontaneous form of arthritis by 13 weeks of age (the latest timepoint in our experiments).

## Histology

For histology, paws were fixed in 4% paraformaldehyde (PFA) overnight at 4 °C and decalcified in 10% EDTA for 3–4 weeks. Decalcified paws were embedded in optimal cutting temperature (OCT) and cryosectioned or embedded in paraffin wax and sectioned before staining with haematoxylin and eosin, or Safranin O, haematoxylin and fast green. Scoring of histological sections was performed blinded using an arthritis severity score as previously described[18].

## Immunofluorescence staining protocol

Frozen tissue sections were thawed for 30 min at room temperature and washed with PBS before permeabilization with 0.25% Triton X-100 or 0.5% saponin in PBS for 10 min. After permeabilization, sections were washed with PBS again and blocked with 3% BSA at room temperature for 30 min. Tissue sections were stained with primary antibodies (anti-cit-H3 (Abcam), anti-DNA/H1 (Merck) and anti-GR-1 (produced in-house[59])) diluted in blocking buffer (3% BSA in PBS) for 1 h 30 min in a humidified chamber, after which they were washed with PBS-Tween-20 (0.05%) and stained with secondary antibodies for 1 h at room temperature in the dark. Sections were washed as before and counterstained with 1 μg ml$^{-1}$ DAPI and mounted in Vectashield antifade mounting medium for fluorescence (Vectorlabs). Coverslips were sealed with nail polish and slides were stored in the dark at 4 °C until imaging. Fluorescence was visualised using the Zeiss confocal LSM700 microscope and Zen Black software (Zeiss). The quantification of positive area (area$^+$ %) for each channel was conducted using Fiji software. Image preprocessing included utilizing the built-in 'Moments' algorithm for thresholding each channel.

## Neutrophil isolation

Bone marrow neutrophils were isolated using Histopaque (Merck) by a density gradient centrifugation method or using the EasySep Mouse Neutrophil Enrichment Kit (StemCell Technologies) according to the manufacturer's guidelines. Flow cytometry analysis confirmed an approximately 90% pure neutrophil population (the remaining population consisting primarily of monocytes). To isolate thioglycollate-elicited neutrophils, mice were injected i.p. with 1 ml of 3% thioglycollate broth. After 4 h, mice were culled and neutrophils were harvested by peritoneal lavage with PBS 5 mM EDTA.

Human neutrophils from the blood of healthy donors were purified using a Ficoll-Paque density centrifugation method[60] or using the EasySep direct human neutrophil isolation kit (StemCell Technologies) as per the manufacturer's instructions. Samples were obtained from consenting healthy donors with the approval of the Faculty of Health and Life Sciences ethics review board, University of Exeter (eCLES-Bio000371) and the College of Life Sciences and Medicine ethics review board, University of Aberdeen (CERB number 1243).

## CD34+ cell isolation and CRISPR-mediated knockout

Peripheral blood mononuclear cells (PBMCs) were isolated from apheresis blood waste (NHSBT) with ethical approval from NHS Research Ethics Committee (REC 18/EE/0265) by density centrifugation using Histopaque-1077 (Sigma-Aldrich) according to the manufacturer's instructions. After washing, cells were resuspended in red cell lysis buffer (55 mM NH$_4$Cl, 0.137 mM EDTA and 1 mM KHCO$_3$, pH 7.5) and incubated on ice for 10 min. Next, to enrich haematopoietic stem cells, CD34$^+$ cells were isolated from PMBCs using a human CD34 Microbead Kit (Miltenyi Biotec) according to the manufacturer's protocol. Isolated cells were cultured in Iscove's modified Dulbecco's medium (IMDM) supplemented with 10% (v/v) FBS and 1% (v/v) penicillin–streptomycin at 37 °C in 5% CO$_2$. Cytokines were added at the indicated concentrations and days of culture: stem cell factor (SCF; 50 ng ml$^{-1}$; day 0–5 of culture), Flt-3 ligand (50 ng ml$^{-1}$; day 0–5 of culture), interleukin-3 (IL-3; 10 ng ml$^{-1}$; day 0–5 of culture), granulocyte–macrophage

colony-stimulating factor (GM-CSF; 10 ng ml$^{-1}$; day 3–7 of culture) and granulocyte colony-stimulating factor (G-CSF; 10 ng m$^{-1}$; day 7–14 of culture). All functional assays were completed between day 17 and day 18 of culture[49].

CRISPR-mediated knockout was completed using the AmaxaTM 4D-Nucleofector (Lonza) using a P3 Primary Cell 4D-NucleofectorTM X Kit S (Lonza) and TrueCut Cas9 Protein v2 (Invitrogen) according to the manufacturers' instructions. In brief, day 3 cultured neutrophils were resuspended in the mixture of P3 Primary Cell Solution and supplement 1 containing 50 pmol Cas9 and 125 pmol of guide RNA (62.5 pmol of two guides with the same gene target). Cells were transferred into nucleocuvette strip and electroporated using the EO-100 program. After electroporation cells were transferred into six-well plates containing StemSpan medium (StemCell Technologies) supplemented with FBS and penicillin–streptomycin and containing SCF, Flt-3 and IL-3. From day 5 onwards, cells were cultured as outlined above in IMDM. All functional assays were completed between day 17 and day 18 of culture. Guide RNAs were designed using Knockout Guide Design (Synthego). The following single guide RNAs were used (Synthego, modified single guide RNA with EZ scaffold): CLEC12A + 9979452: UGAAUAUCUC CAACAAGAUC and CLEC12A-9979406: GUUGUAGAGAAAUAUUUCUC; negative control scrambled #1: GCACUACCAGAGCUAACUCA and negative control scrambled #2: GUACGUCGGUAUAACUCCUC. Cells were stained with anti-CD66-PE/Dazzle (clone G10F5; diluted 1:50), anti-CD15-AF700 (clone Hi98; diluted 1:50), anti-CD16-APC (clone 3G8; diluted 1:50) and anti-hMICL or isotype control antibodies at 10 µg ml$^{-1}$ to confirm loss of MICL expression.

## ROS

Bone marrow neutrophils and human neutrophils were isolated as described above and resuspended in OptiMEM (Thermo Fisher Scientific) supplemented with 5% FCS. Cells ($5 \times 10^5$) were added to each well of a white 96-well flat-bottomed plate and stimulated in triplicate with 200 µg ml$^{-1}$ MSU crystals (Invivogen), 25 µg ml$^{-1}$ Zymosan (Molecular Probes), 100 nM PMA (Sigma), isolated NETs (100 µg ml$^{-1}$ based on NET–DNA concentration) or *A. fumigatus* hyphae ($1 \times 10^4$ conidia per well incubated for 12 h at 37 °C) in the presence of 100 µM luminol (Sigma). Chemiluminescence was measured every 3 min for 2 h in a FLUOStar Optima microplate reader (BMG Labtech) or a Spark Cyto (Tecan) at 37 °C with 5% CO$_2$.

## NETs

Bone marrow and thioglycollate-elicited neutrophils were isolated as described above. Cells were resuspended in RPMI (without phenol red; Thermo Fisher Scientific) supplemented with 2% DNase$^{-/-}$ mouse serum and seeded in an eight-well iBidi µ-slide (iBidi). Cells were stimulated with 100 µg ml$^{-1}$ of MSU crystals (Invivogen), 100 nM PMA (Sigma), isolated NETs (100 µg ml$^{-1}$ based on NET–DNA concentration) or *A. fumigatus* hyphae and incubated at 37 °C with 5% CO$_2$ for 4 h. In some experiments, as indicated in the text, DPI (10 µM; Sigma) or NSC-87877 (5 µM; Cayman Chemical) were included in the assays. Extracellular DNA was visualized with 5 µM Sytox Green (Invitrogen). In experiments without fixation, cells were previously stained with Draq5 (Thermo Fisher Scientific) and cell-impermeable Sytox Green, and images were acquired on an inverted Zeiss AxioObserver Z1 using a PlanApo ×20/0.75 NA dry lens (Carl Zeiss) and a Hamamatsu Fusion sCMOS camera with an attached incubation chamber (PeCon GmbH) at 37 °C. Fluorescent images were analysed in Image J, and NETs defined as Sytox Green-positive cells showing extrusions were counted.

NET (%) = (total Sytox Green-positive cells extruding NETs/total cells counted) × 100

For kinetic curves of NET formation, neutrophils were seeded in a 96-well black plate with a transparent bottom and the cells were left to adhere for 30 min in a cell culture incubator. Cells were stimulated with the defined reagents and 5 µM Sytox Green. Fluorescence signal (504-nm excitation and 523-nm emission) was measured every 10 min for 7 h in a Spark Cyto (Tecan) at 37 °C with 5% CO$_2$.

NETs were isolated as previously described[61]. In brief, bone marrow-isolated or purified human neutrophils were plated in a six-well plate at a density of $1 \times 10^6$ cells per well in RPMI without phenol red. Following stimulation with 100 nM PMA for 4 h at 37 °C, the culture medium was removed and NETs were partially digested by application of a restriction enzyme mix combining the enzymes BseRI, PacI, NdeI and AfIII (New England Biolabs) at a concentration of 5 U ml$^{-1}$ in NEB buffer for 1 h at 37 °C. Supernatants were collected and centrifuged at 300$g$ for 10 min at 4 °C. NET supernatants were transferred to a fresh tube and stored at −80 °C until used. When indicated, NETs were treated with DNase I (Thermo Fisher Scientific) or proteinase K (Roche) for 1 h at 37 °C.

Recognition of isolated NETs by MICL was assessed by ELISA. A 96-well plate (Nunc Maxisorp) was coated with NETs diluted in PBS overnight at 4 °C. Wells were washed and blocked with blocking buffer (5% BSA) at room temperature. Fc–mMICL or Fc–mCLEC12B at 1 µg ml$^{-1}$ in blocking buffer was added and incubated for 2 h at room temperature. Bound Fc-fusion proteins were detected with horseradish peroxidase-conjugated goat anti-human IgG (Jackson Immunoresearch) diluted 1:10,000 for 1 h. TMB substrate was added, and absorbance was measured using a plate reader (Tecan). Purification of Fc–MICL and Fc–mCLEC12B, from transduced HEK293T cells (originally purchased from the American Type Culture Collection (ATCC), but not tested for *Mycoplasma* contamination for the experiments detailed in this paper), was performed as previously described[32].

To analyse the recognition of isolated NETs by MICL using a MICL-expressing cell line, a 96-well plate was coated with NETs diluted in PBS overnight at 4 °C. Wells were washed, and BWZ.36 NFAT-LacZ cells (provided by W. Yokoyama; not tested for *Mycoplasma* contamination for the experiments detailed in this paper) expressing the CD3ζ chain fused to the transmembrane and carbohydrate recognition domain (CRD) of mMICL or mCLEC12B were added ($2 \times 10^5$ cells per well) for 18 h at 37 °C. After stimulation, cells were centrifuged at 800$g$ for 2 min and washed two times with PBS. Of CPRG substrate buffer, 100 µl was added per well. The reaction was stopped 4 h later with the addition of 50 µl of glycine-EDTA buffer, and the absorbance was measured using a plate reader[62] (Tecan). Antibody crosslinking with the appropriate receptor antibody was used to confirm the functionality of the chimeric receptor constructs.

To analyse neutrophil infiltration by isolated NETs, 6–8-week-old *Micl*$^{-/-}$ and C57BL/6J mice were injected i.p. with 300 µg NETs or 50 µg LPS. Four hours later, mice were euthanized, and their peritoneal cells were counted and analysed by flow cytometry. Cell-free DNA was evaluated using the Quant-iT PicoGreen dsDNA Assay (Invitrogen) following the manufacturer's instructions.

## Immunofluorescent staining of in vitro samples

Neutrophils ($5 \times 10^4$) were plated in an eight-well iBidi µ-slide (iBidi) 1 h before stimulation with isolated NETs, PMA or MSU. Four hours after stimulation, cells were fixed for 20 min at 37 °C with 2% PFA and permeabilized with 0.5% Triton X-100 in PBS. Samples were blocked with 3% (v/v) normal goat serum, 1% (w/v) BSA in PBS and incubated with anti-cit-H3 (Abcam) and anti-myeloperoxidase (R&D) antibodies. The secondary antibodies donkey anti-rabbit Cy5 and rabbit anti-goat Alexa Fluor 488 were used. Finally, the samples were stained with DAPI.

Fluorescence quantification was conducted by segmenting the DAPI channel using QuPath software (version 0.4.3) with the Cellpose extension (https://github.com/BIOP/qupath-extension-cellpose). Before segmentation, preprocessing of the DAPI channel was carried out using Fiji software to mitigate background noise, involving background subtraction and enhanced contrast built-in functions.

Preprocessing of the GFP and Texas Red channels involved a background subtraction of the mean grey value and two times the standard deviation of an empty region (background noise). The quantification of each channel was conducted using the measurement function of Fiji software, utilizing the mask generated by Qupath.

## Flow cytometry and monoclonal antibodies

Cells were isolated from the hind paw ankle joint of arthritic mice using the protocol previously described[20]. Peripheral blood was collected by cardiac puncture or tail nicking in the presence of EDTA and red blood cell lysis performed using PharmLyse (BD Biosciences). Single-cell suspensions were stained with fixable viability dye eFluor 780 (eBioscience) and further stained with conjugated antibodies for same-day acquisition or fixed in 2% PFA. Conjugated antibodies used in these experiments included: anti-CD45-FITC (clone 102), anti-CD45-PerCP-cyanine5.5 (clone 102), anti-CD11b-BUV395 (clone M1/70), anti-CD11b-PE-Cy7 (clone M1/70), anti-GR-1-APC (clone RB6-8C5), anti-MHC-II-FITC (clone 2G9), anti-MHC-II-BUV496 (clone 2G9), anti-C5aR-BV510 (clone 20/70), anti-Ly6G-BV421 (clone 1A8), anti-Ly6G-Spark Blue 550 (clone 1A8), anti-Ly6G-APC (clone 1A8), anti-CD62L-BV510 (clone MEL-14), anti-CD18-BV650 (clone C71/16), anti-CD18-APC (H155-78), anti-F4/80-AF700 (clone BM8), anti-F4/80-PE-Cy7 (clone BM8), anti-Ly6C-PE-Cy7 (clone HK1.4), anti-Ly6C-Brilliant Violet 570 (clone HK1.4), anti-CCR1-PE (clone 643854), anti-CXCR2-APC (clone SA045E1), anti-CD11c-BV711 (clone HL3), anti-B220-BV605 (clone RA3-6B2) and anti-CD3-Alexa Fluor 647 (clone 17A2). All were purchased commercially from eBioscience, R&D systems or BioLegend and used diluted 1:600. Anti-mCLEC12A-biotinylated[21], anti-hCLEC12A[63] and isotype control AFRC MAC 49 (ECACC 85060404; isotype for anti-MICL) were generated in-house and used at 10 µg ml$^{-1}$. Anti-mCLEC12A-biotinylated[21] and anti-hCLEC12A[63] antibodies were validated using mouse and human CLEC12A-transduced NIH3T3 fibroblast (originally purchased from the ATCC; not tested for *Mycoplasma* contamination for the experiments detailed in this paper). Cells were acquired using a BD LSR II Fortessa flow cytometer (BD Biosciences), BD LSRFortessa X-20 flow cytometer (BD Biosciences) or Cytek Aurora Spectral cytometer (Cytek). See Extended Data Fig. 1b for the cellular gating strategy. Data were collected using BD FACSDiva v8.0.3 (BD Biosciences) or SpectroFlo software v3.2.1 (Cytek) and analysed using FlowJo v10 software (BD Biosciences).

## Imaging flow cytometry

Single cells isolated from the arthritic joint, as detailed above, were fixed in 1% PFA for 30 min at 37 °C. Cells were washed with PBS 2 mM EDTA for 5 min at 500g and stained with conjugated antibodies (detailed above), anti-DNA/histone 1 (Merck Millipore; 1.4 µg ml$^{-1}$) and anti-cit-H3 (Abcam; diluted 1:300) for 1 h at room temperature. The secondary antibodies goat anti-rabbit APC (Molecular probes) and goat anti-mouse AF488 (Invitrogen) at 1 µg ml$^{-1}$ were added for 1 h at 4 °C. Cells were washed and resuspended in 50 µl of PBS 2 mM EDTA before acquisition using the Amnis ImageStreamX MKII (Luminex) INSPIRE acquisition software. Files were analysed using IDEAS software v6.2 (Luminex).

Single cells were selected by plotting the area feature of brightfield channel 1 (BF1) versus the aspect ratio parameter of the same brightfield channel, which is the ratio of minor axis to the major axis of the applied mask, and describes the shape of the mask applied to the cells (Extended Data Fig. 7f). Focused cells were then selected by plotting the 'Gradient RMS' feature of BF1 against the BF1 contrast parameter. Cells with high-gradient and high-contrast value were more in focus and chosen for further analysis. All focused cells were used in the analysis for the presence of the cellular fluorescence parameters. Fluorescence parameters were measured in channel 2 (DNA/histone H1-AF488), channel 11 (cit-H3-APC) and channel 7 (Ly6G-BV421) with magnification set to ×40.

## Serum collection from patients with rheumatoid arthritis

Serum samples from patients with rheumatoid arthritis were obtained from the SERA cohort[42] (Extended Data Table 1). Sera from healthy controls were obtained by collection of whole blood in a BD vacutainer serum collection tube (BD Biosciences). The blood was allowed to clot for 15–30 min at room temperature, after which the tubes were centrifuged at 2,000g for 10 min in a 4 °C. Sera were removed, aliquoted and stored at −80 °C. Serum samples were obtained from consenting healthy donors with the approval of the College of Life Sciences and Medicine ethics review board, University of Aberdeen (CERB number 1243).

## Serum collection from patients with COVID-19

Blood from 121 patients diagnosed with SARS-CoV-2 infection at Weill-Cornell Medicine between March and July 2020. Research on patients with COVID-19 was reviewed and approved by the Institutional Review Board of Weill-Cornell Medicine (New York Presbyterian and Lower Manhattan hospitals; #IRB 20-03021645 and #IRB 20-03021671). Informed consents were obtained from all enrolled patients and health-care workers by trained staff, and records were maintained in our research database for the duration of the study. All patients were classified as mild/moderate (*n* = 25) and severe (*n* = 66) disease according to oxygen requirements with mild/moderate disease defined as SARS-CoV-2 infection and less than 6 l of non-invasive supplementary oxygen to maintain SpO2 > 92%, and severe disease was defined as SARS-CoV-2 infection requiring hospitalization and received 6 l or more supplementary oxygen or mechanical ventilation. For controls, we used blood samples from 36 SARS-CoV-2-negative individuals collected by the JRI IBD Live Cell Bank Consortium at Weill-Cornell Medicine. Heparinized plasma and serum samples were aliquoted, heat-inactivated at 56 °C for 1 h and then stored at −80 °C.

## Serum collection from patients with SLE

All patients with SLE in the study met the revised American College of Rheumatology criteria[64] and the SLICC criteria[65]. Some patients had a history of biopsy-proven nephritis according to the International Society of Nephrology/Renal Pathology Society classification. Healthy female volunteers (with no family history of autoimmune disease) served as age-matched and ethnicity-matched controls. All patients provided informed consent, and samples used in this research project were obtained from the Imperial College Healthcare Tissue Bank (ICHTB). The ICHTB is supported by the National Institute for Health Research (NIHR) Biomedical Research Centre based at Imperial College Healthcare NHS Trust and Imperial College London. The ICHTB is approved by Wales REC3 to release human material for research (22/WA/0214), and the samples for this project (ref: R13010a) were issued from sub-collection reference number IMM_MB_13_001.

## ELISAs from human patients

Sera from patients with rheumatoid arthritis, COVID-19 and SLE and healthy controls were screened for the presence of MICL autoantibodies using a modified ELISA method[25]. In brief, Nunc Maxisorp 96-well plates were coated with equivalent amounts of the Fc-fusion proteins Fc–hMICL and Fc–mCLEC12B diluted in PBS overnight at 4 °C. Plates were blocked with 10% BSA and 10% HI goat serum (Merck) in PBS for 1 h. Sera from patients with rheumatoid arthritis, COVID-19 and SLE and healthy control were diluted to a 1:256 dilution in PBS and added to the pre-blocked plate and incubated for 2 h. Bound autoantibodies were detected with horseradish peroxidase-conjugated goat anti-human F(ab')2 fragment (Jackson Immunoresearch) diluted 1:50,000 in PBS for 1 h. TMB substrate was added, and absorbance was measured at 450 nm in a plate reader (Tecan).

### *A. fumigatus* systemic infection model

$Micl^{-/-}$ and C57BL/6J female mice were injected intravenously with $10^6$ *A. fumigatus* ATCC 13073 conidia, as previously described[66]. Mice were culled when they lost 20% body weight or had become moribund. To investigate the role of NET formation, mice were injected i.p. with 4 mg kg$^{-1}$ daily of GSK484 (Cambridge Biosciences) from day 1 to day 5. Organs were homogenized in PBS and used for the determination of fungal burdens and levels of inflammatory cytokines. Fungal burdens were determined by serial dilution onto potato dextrose agar plates and normalized to organ weights. Cytokines were measured by ELISA (BD Biosciences), as described by the manufacturer, and normalized to protein concentration.

### Statistical analysis

Data are represented as mean ± s.d., unless otherwise indicated. All statistical analyses were performed using GraphPad Prism (v9, Graph-Pad Software) and depicted in the respective figure legends. For all experiments with two groups, two-tailed unpaired Student's *t*-tests (equal variances) or two-tailed Mann–Whitney tests were used. One-way or two-way ANOVA (with equal variances) with correction for multiple comparisons was performed for experiments with more than two groups. All $P < 0.05$ were considered statistically significant.

### Reporting summary

Further information on research design is available in the Nature Portfolio Reporting Summary linked to this article.

### Data availability

All data necessary for the conclusions of this study are provided with the paper. Additional data on patients with rheumatoid arthritis are available on request from the SERA and approval by the SERA Access Committee[42]. Source data are provided with this paper.

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

**Acknowledgements** We thank the staff of the animal facilities at the University of Aberdeen and the University of Exeter for support and care for animals; C. Paterson from the University of Glasgow for assistance in establishing a Material Transfer Agreement; C. Parkin and D. Thompson for support with microscopy; and M. Stacey for valuable input. We acknowledge funding from the Wellcome Trust (102705 and 097377), Versus Arthritis (21164, 20775 and 21156), the US National Institutes of Health (R01DK121977 and R01AI163007), Versus Arthritis Centre of Excellence, Medical Research Council (MR/L020211/1) and the MRC Centre for Medical Mycology (MR/N006364/1). SLE tissue samples were provided by the Imperial College Healthcare Tissue Bank funded by the National Institute for Health Research (NIHR), Biomedical Research Centre based at the Imperial College Healthcare NHS Trust and Imperial College London. The views expressed are those of the authors and not necessarily those of the NHS, the NIHR or the Department of Health.

**Author contributions** G.D.B., J.A.W., M.M. and L.W. conceived and designed the study and guided the interpretation of the results. M.M. and L.W. performed the majority of the experiments and data analysis. F.C., A.J.R., I.M.D., A.P.-B., F.S., R.S., E.A.S., C.R., P.L.M., P.R., D.S., R.Y. and D.M.R. assisted with experiments. P.Z. performed CRISPR-mediated knockout of haematopoietic stem cells. F.P. and T.B. performed microscopy experiments and analysis. J.H. provided support with animal experiments. A.McIntosh, A.McConnachie, T.K., I.D.I., I.B.M., M.B. and M.C.P. provided the human patient data and analysis. C.D.B., M.J.G.F. and B.A. provided critical conceptual input and reagents. R.A. and D.R.B. provided reagents. M.M. and G.D.B. wrote the manuscript. All of the authors provided comments on and approved the final version of the manuscript.

**Competing interests** The authors declare no competing interests.

**Additional information**
**Correspondence and requests for materials** should be addressed to Gordon D. Brown.

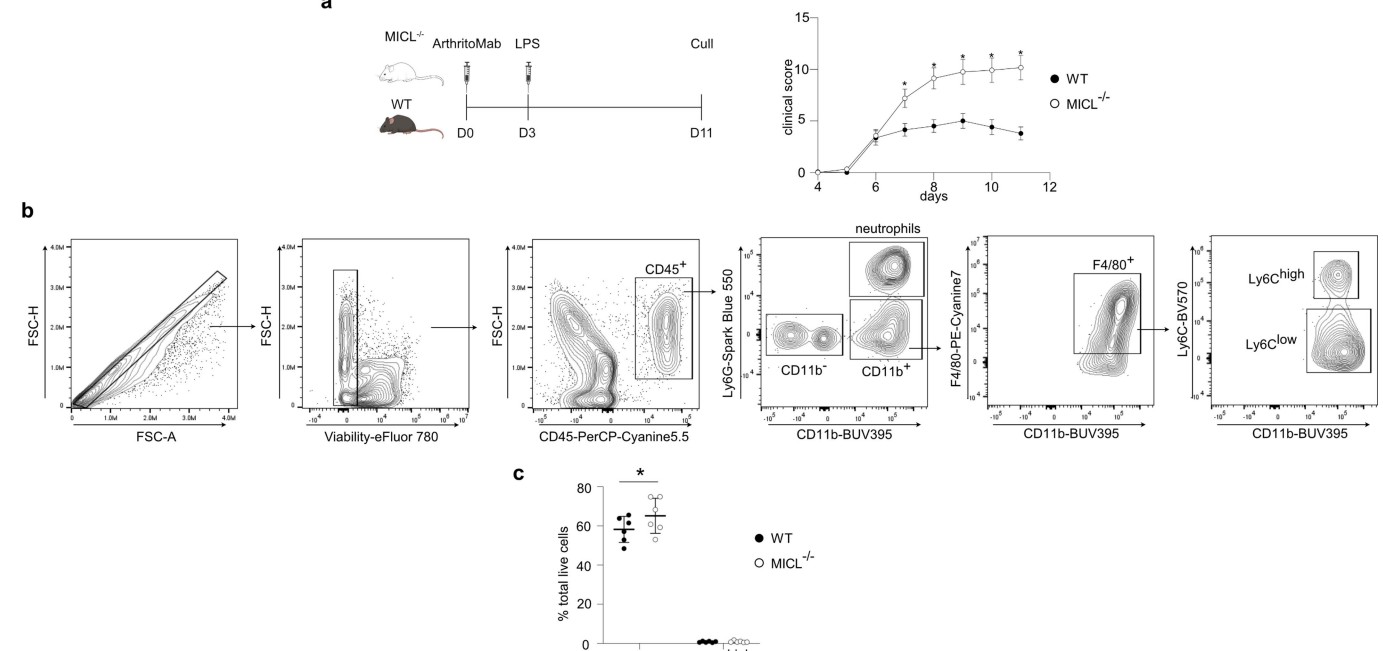

**Extended Data Fig. 1 | MICL regulates inflammation during arthritis.**
**a**, Schematic of the CAIA model and severity scoring shown as mean ± SEM (pooled data from three independent experiments, n = 16 biologically independent mice) assessed over time. Statistical significance was determined by two-way ANOVA with Bonferroni's multiple comparisons test. Schematic in panel **a** was created using BioRender (https://biorender.com). **b**, Flow cytometry gating strategy used for characterisation of the ankle joint cellular infiltrate. Cells isolated from the inflamed joint were first gated to identify single viable cells using a fixable viability dye. CD11b+ populations were further gated into subsets namely; neutrophils (Ly6G+CD11b+) and Ly6C high cells (Ly6C high F4/80+CD11b+) or Ly6C low cells (Ly6C low F4/80+CD11b+). Antigen presenting cells (APCs) were gated as MHC II+CD11b+. **c**, Neutrophils and Ly6C high (defined as shown in the gating strategy in Extended Data Fig. 1b) in the ankle joint during CAIA at day 5 are represented as a percentage of total live cells (n = 1 experiment with 6 mice/group). Statistical significance determined by two-way ANOVA with Bonferroni's multiple comparisons test. WT, wild-type mice; *p < 0.05.

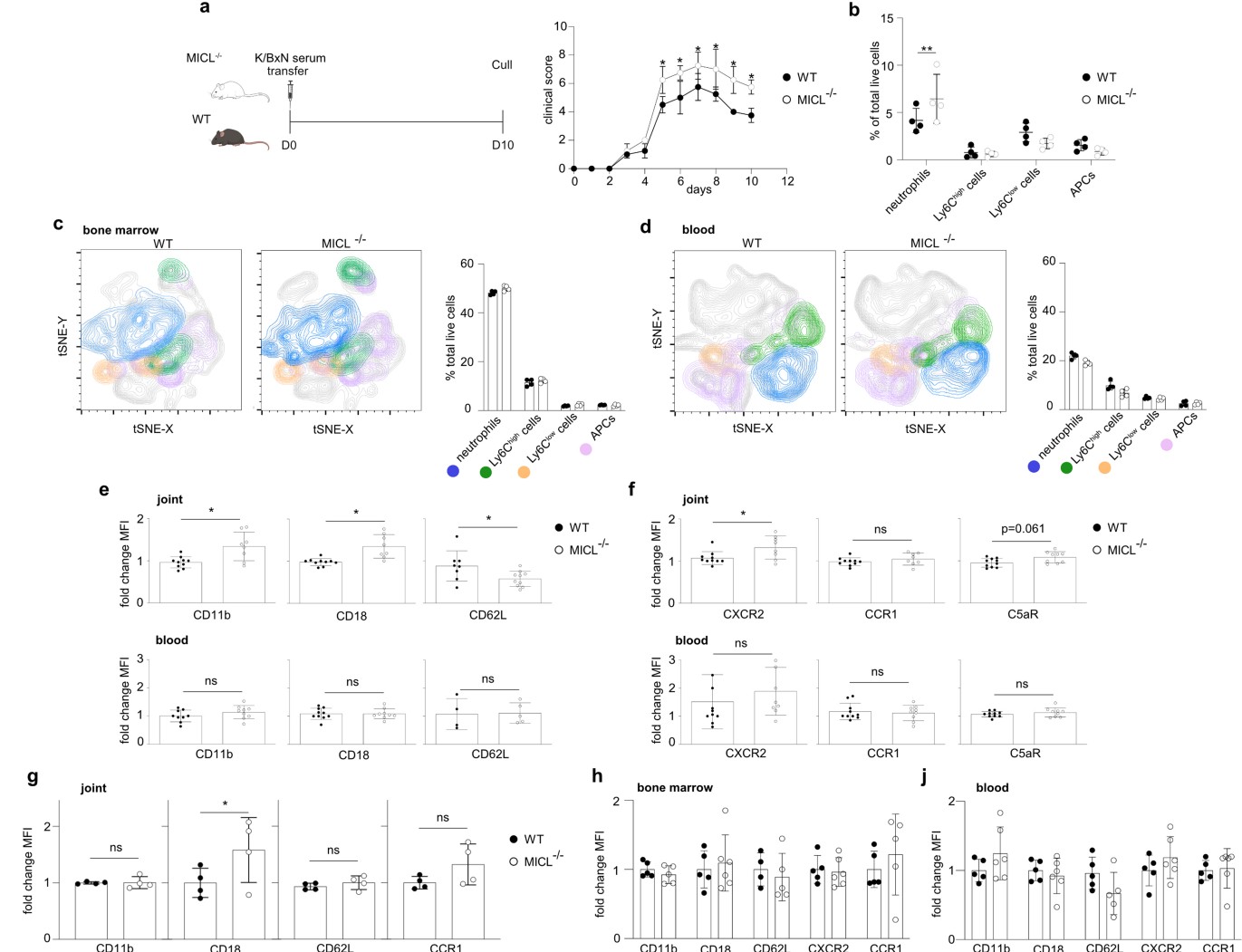

**Extended Data Fig. 2 | MICL is required for controling neutrophil responses during arthritis. a**, Schematic of K/BxN serum transfer model and severity scoring shown as mean ± SD analysed over time. **b**, Total live cell populations (defined by gating strategy in Extended Data Fig. 1b) isolated from inflamed joints on day 10, mean ± SD. Neutrophils WT vs MICL⁻/⁻ p = 0.0086. **a**,**b**, Data is a representative example of three independent experiments, with 4 mice/group/experiment, and analysed using two-way ANOVA with Bonferroni's multiple comparisons test. Schematic in panel **a** was created using BioRender (https://biorender.com). **c**,**d**, tSNE plots of CD45⁺ myeloid populations displayed as neutrophils (blue), Ly6Chigh cells (green), Ly6Clow cells (orange), antigen presenting cells (APCs) (pale violet), and all other subpopulation (grey) in the bone marrow (**c**), and blood (**d**) of naïve mice. Data is a representative example of three independent experiments, with 4 mice/group/experiment, represented as mean ± SD. **e**,**f**, Fold change in mean fluorescent intensity (MFI), relative to

wild-type mice of (**e**) neutrophil activation markers (CD11b, CD18, CD62L) and (**f**) chemokine receptors (CXCR2, CCR1, C5aR), isolated concomitantly from joints or blood of arthritic mice during CAIA on day 7 (pooled data from two independent, n = 10 biologically independent mice). **g**, Fold change in mean fluorescence intensity (MFI) on neutrophils, relative to wild-type mice of CD11b, CD18, CD62L and CCR1, isolated from the joints of arthritic mice on day 10. Data is a representative example of n = 3 independent experiments with 4 mice/group/experiment, shown as mean ± SD and analysed using two-way ANOVA with Bonferroni's multiple comparisons test. **h**,**j**, Fold change in mean fluorescent intensity (MFI) relative to wild-type mice of neutrophil activation markers (CD11b, CD18, CD62L) and chemokine receptors (CXCR2, CCR1), isolated concomitantly from bone marrow (**h**) and blood (**j**) of naïve mice (representative example of n = 2 experiments with 5 mice/group/experiment). WT, wild-type mice; KO, MICL⁻/⁻ mice; *p < 0.05; ns, not significant.

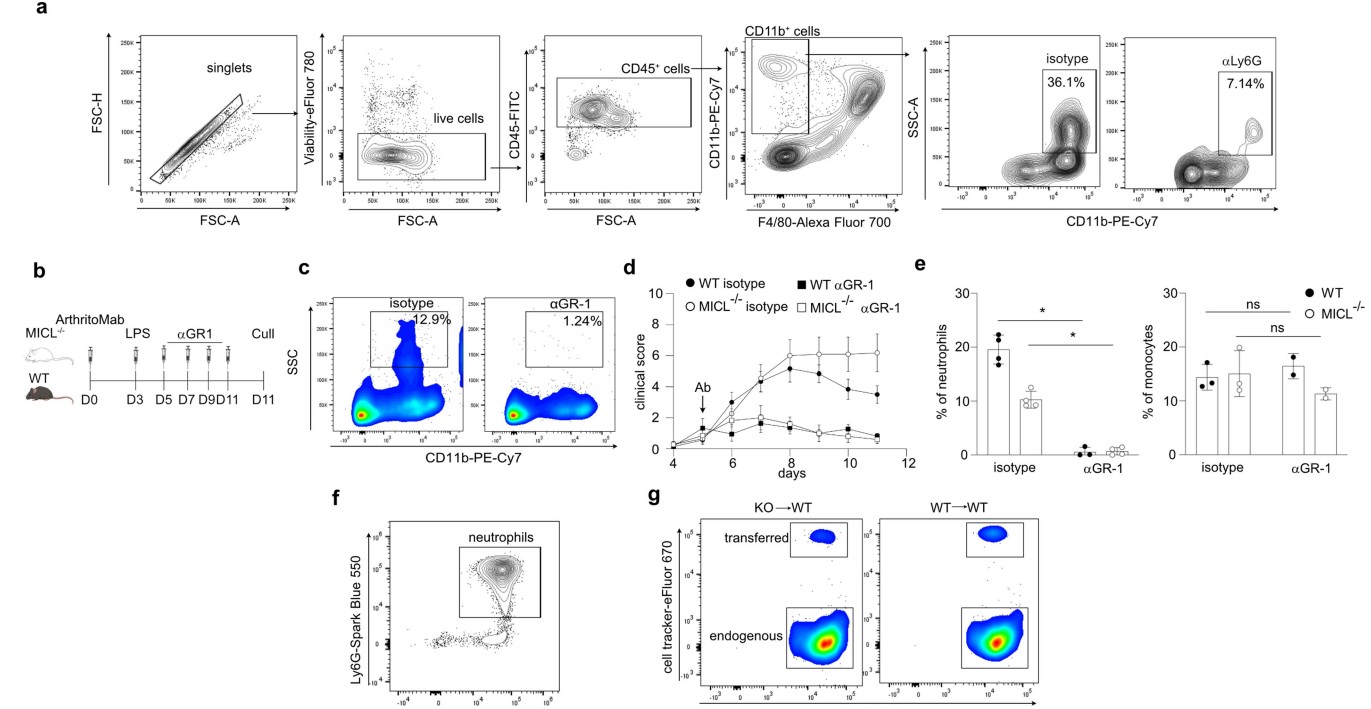

**Extended Data Fig. 3 | Neutrophil depletion reduces inflammation in WT and MICL$^{-/-}$ mice during arthritis. a**, Flow cytometry gating strategy used to confirm αLy6G antibody-mediated neutrophil depletion in CAIA model at day 9. Neutrophils were identified as CD45$^+$CD11b$^+$SSC$^{high}$ cells. **b**, Schematic representation of the αGR-1-mediated neutrophil depletion strategy in the CAIA model. Schematic in panel **b** was created using BioRender (https://biorender.com). **c**, Flow cytometry contour plot-illustrating depletion of peripheral blood neutrophils 48 h post-injection with αGR-1. **d**, Severity scoring of WT and KO mice treated with isotype or αGR-1 antibodies, as

indicated (pooled data from two independent experiments, n = 12 biologically independent mice). Data is represented as mean ± SEM. **e**, Quantification of neutrophils (CD45$^+$CD11b$^+$SSC$^{hi}$) and monocytes (CD45$^+$CD11b$^+$F4/80$^+$) in the peripheral blood on day 11. $n$ = 4 (neutrophils) or $n$ = 3 (monocytes) biologically independent mice represented as mean ± SD. **f**, Contour plot showing purity of neutrophils after bone marrow isolation. **g**, Pseudo-colour plot illustrating endogenous and cell-tracker labelled, adoptively transferred CD45$^+$CD11b$^+$Ly6G$^+$ neutrophils in the joint at day 9 (48 h post-transfer). WT, wild-type mice; KO, MICL$^{-/-}$ mice; Ab, αGR-1 or isotype; ns, not significant; *, $p$ < 0.05.

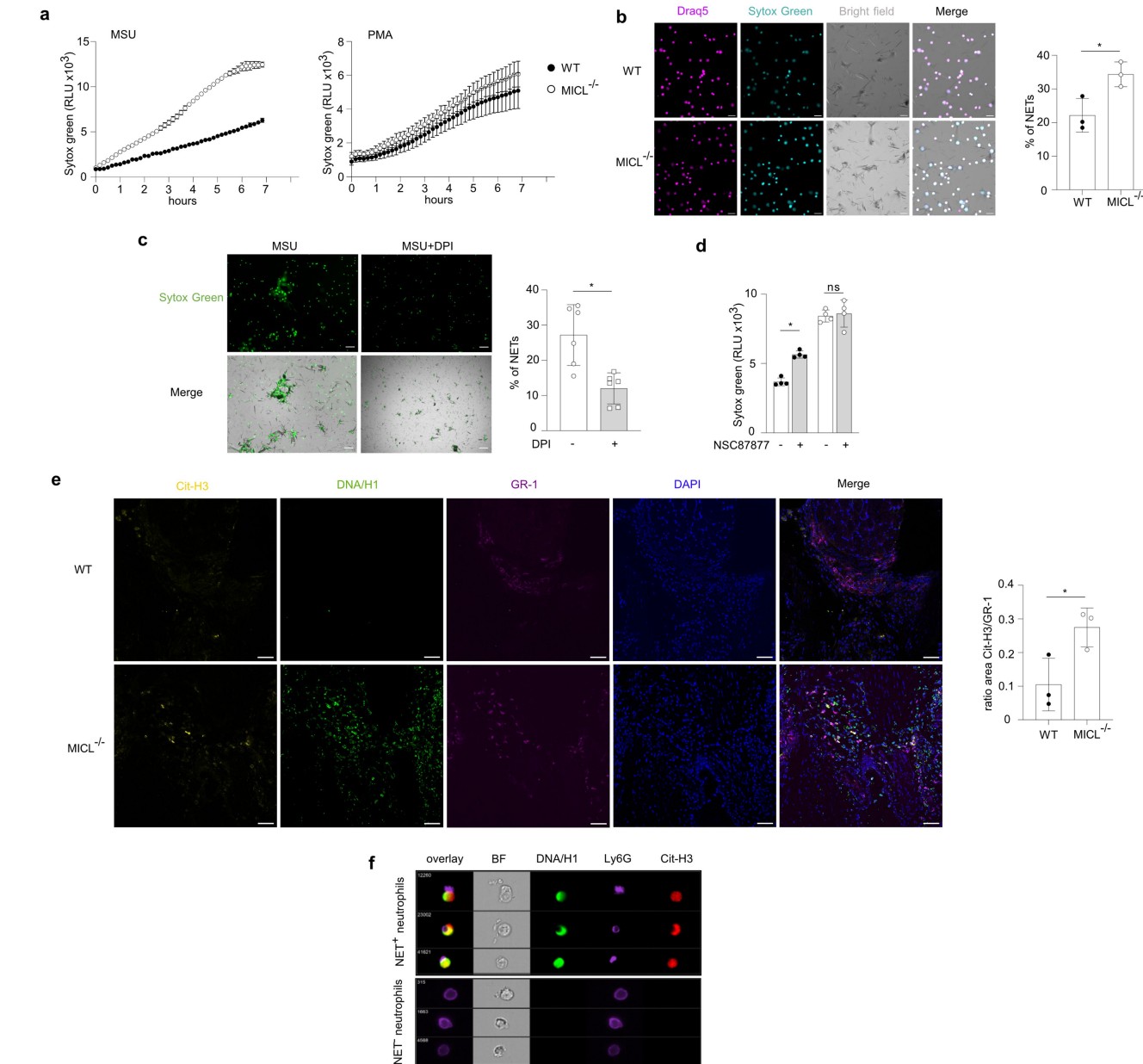

**Extended Data Fig. 4 | MICL regulates NET formation. a**, Fluorescence of NET-bound Sytox green of MSU or PMA-induced NETs in thioglycollate-elicited neutrophils analysed using a SPARK CYTO reader (Tecan) every 10 min for up to 7 h. Data is a representative example of n = 3 independent experiments, mean ± SD performed in triplicate. **b**, Representative Sytox green fluorescent (cyan), Draq5 (magenta), and bright field (grey) images of MSU-induced NETs in bone marrow neutrophils isolated from WT and MICL-deficient animals. Cells were stained unfixed. Quantification (as percentage of Sytox green positive cells extruding NETs) shown as mean ± SD, n = 1 experiment performed in triplicate. Statistical significance determined by Student's t-test. **c**, Representative Sytox green fluorescent and bright field images of MSU-induced NETs in thioglycollate-elicited neutrophils isolated from MICL-deficient animals with and without DPI pre-treatment. Scale bar = 100 μm. Quantification (mean ± SD) shown right (n = 2 experiments performed in triplicate). Statistical significance determined by Student's t-test. **d**, Fluorescence of NET-bound Sytox of MSU-induced NETs in thioglycollate-elicited neutrophils with and without NSC87877 treatment analysed 4 h after stimulation. Pooled data from two independent experiments performed in duplicate represented as mean ± SD. **e**, Representative confocal immunofluorescence microscopy images of NETs in CAIA WT and MICL$^{-/-}$ synovial sections on day 11 (as per schedule in Extended Data Fig. 1a). GR-1 (purple), DAPI (blue), Cit-H3 (yellow), and DNA/H1 (green) and quantification of cit-H3/GR-1 area (n = 1 experiment with 3 mice/group). NETs are defined as GR-1+Cit-H3 + DNA/H1+ stained cells. Scale bar = 100 μm. Statistical significance determined by Student's t-test. **f**, Representative imaging flow cytometry examples of NET positive (+) and NET negative (−) neutrophils isolated from the ankle joints of mice at day 11 during CAIA. BF, Brightfield; DNA/H1 (green); Ly6G (purple); Cit-H3 (red). *, *p* < 0.05.

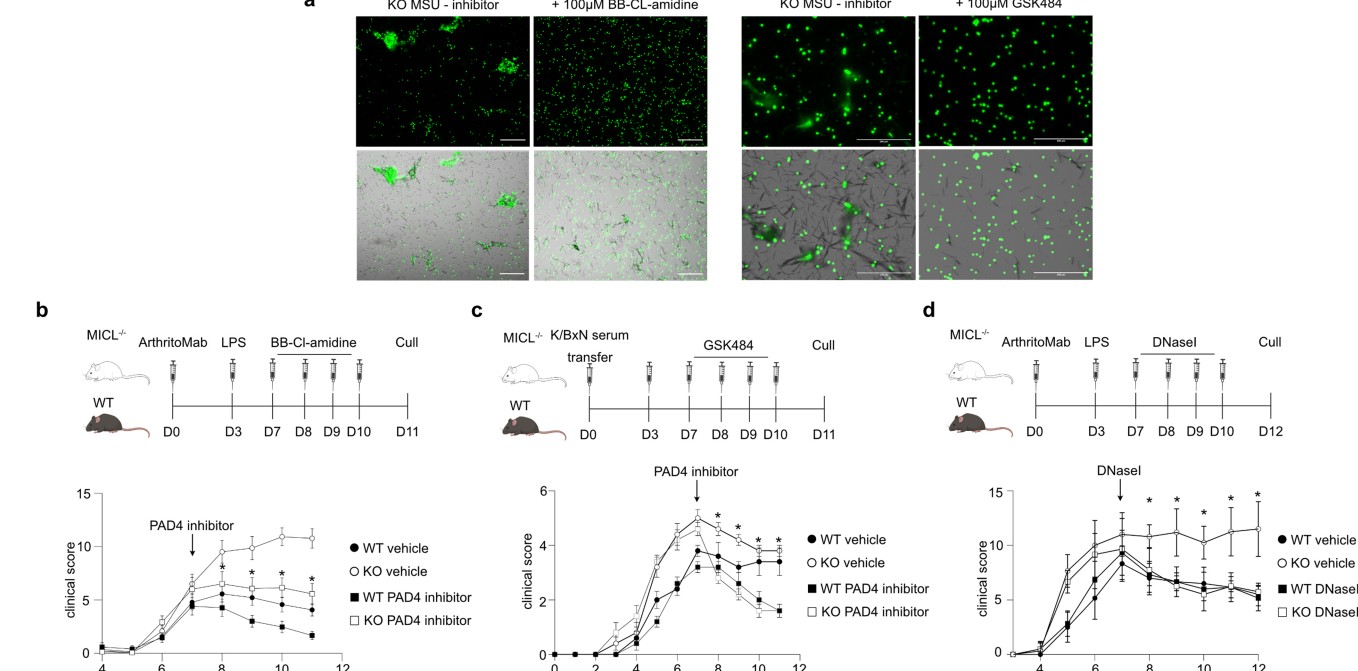

**Extended Data Fig. 5 | Neutrophil NET formation drives arthritis disease severity in MICL⁻ᐟ⁻ mice. a**, Representative Sytox green fluorescent and bright field images of MSU-induced NETs in thioglycollate-elicited neutrophils in the presence of 100 µM BB-Cl-amidine or 10 µM GSK484. Scale bar = 200 µm. **b**, Schematic representation of BB-Cl-amidine treatment regime during CAIA and severity scoring of WT and KO mice during CAIA treated with vehicle or PAD4 inhibitor. Black arrow indicates start of the treatment. Data are represented as mean ± SEM (pooled data from two independent experiments with 5 mice/group/experiment). **c**, Schematic representation of GSK484 treatment regime during K/BxN serum transfer model and severity scoring of WT and KO mice

during K/BxN serum transfer model treated with vehicle or GSK484. Black arrow indicates start of the treatment. Data is a representative example of n = 2 independent experiments, mean ± SD, 5 mice/group/experiment. **d**, Schematic representation of DNaseI treatment regime and severity scoring of WT and KO mice during CAIA (n = 1 experiment with 6 mice/group). Black arrow indicates start of the treatment. **b**,**c**,**d**, Statistical significance was determined by two-way ANOVA with Tukey's *post hoc* test. WT, wild-type mice; KO, MICL⁻ᐟ⁻ mice; ns, not significant; *, p < 0.05. Schematics in panels **b**–**d** were created using BioRender (https://biorender.com).

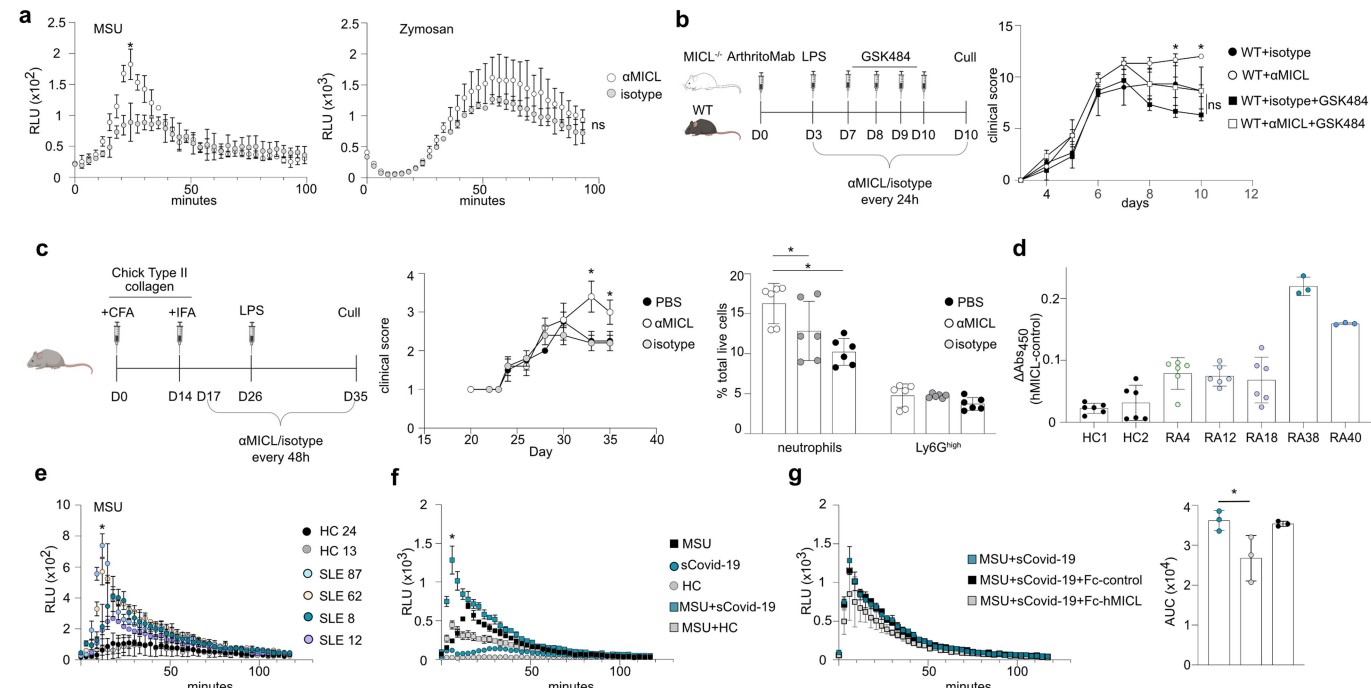

**Extended Data Fig. 6 | Anti-MICL antibodies exacerbate autoinflammatory diseases. a**, ROS generation by MSU (left) or zymosan (right) of neutrophils derived from WT animals in the presence/absence of antibodies targeting MICL (αMICL) depicted as relative light units (RLU) over time. Data is a representative example of n = 2 independent experiments, mean ± SD performed in triplicate. **b**, Schematic and severity scoring of the GSK484 treatment regime during CAIA in the presence of anti-MICL or isotype control monoclonal antibodies. Severity scoring is shown as mean ± SD, n = 1 experiment with 3 mice/group. **c**, Schematic, severity scoring and cell recruitment in the inflamed joints at day 35 during CIA model. Severity scoring is shown as mean ± SD, n = 1 experiment with 6 mice/group, myeloid cell populations are represented as a percentage of total live cells. Statistical significance determined by two-way ANOVA with Tukey's *post hoc* test. Schematics in panels **b**,**c** were created using BioRender (https://biorender.com). **d**, Level of anti-MICL

autoantibodies in serum from healthy controls (HC) and RA patients used to stimulate human neutrophils in Fig. 3c. **e**, ROS generation by MSU of human neutrophils in the presence of serum from SLE patients and healthy controls (HC). Data is a representative example of n = 2 independent experiments, mean ± SD performed in triplicate. **f**, ROS generation by MSU of human neutrophils in the presence of pooled serum from sCOVID-19 patients and healthy controls (HC). Data is a representative example of n = 2 independent experiments, mean ± SD performed in triplicate. **g**, ROS generation and area under curve (AUC) of human neutrophils stimulated with MSU in the presence of pooled serum from sCOVID-19 patients pre-incubated with Fc-hMICL or Fc-control. Data is a representative example of n = 2 independent experiments, mean ± SD performed in triplicate. WT, wild-type mice; KO, MICL[−/−] mice; ns, not significant; *, p < 0.05.

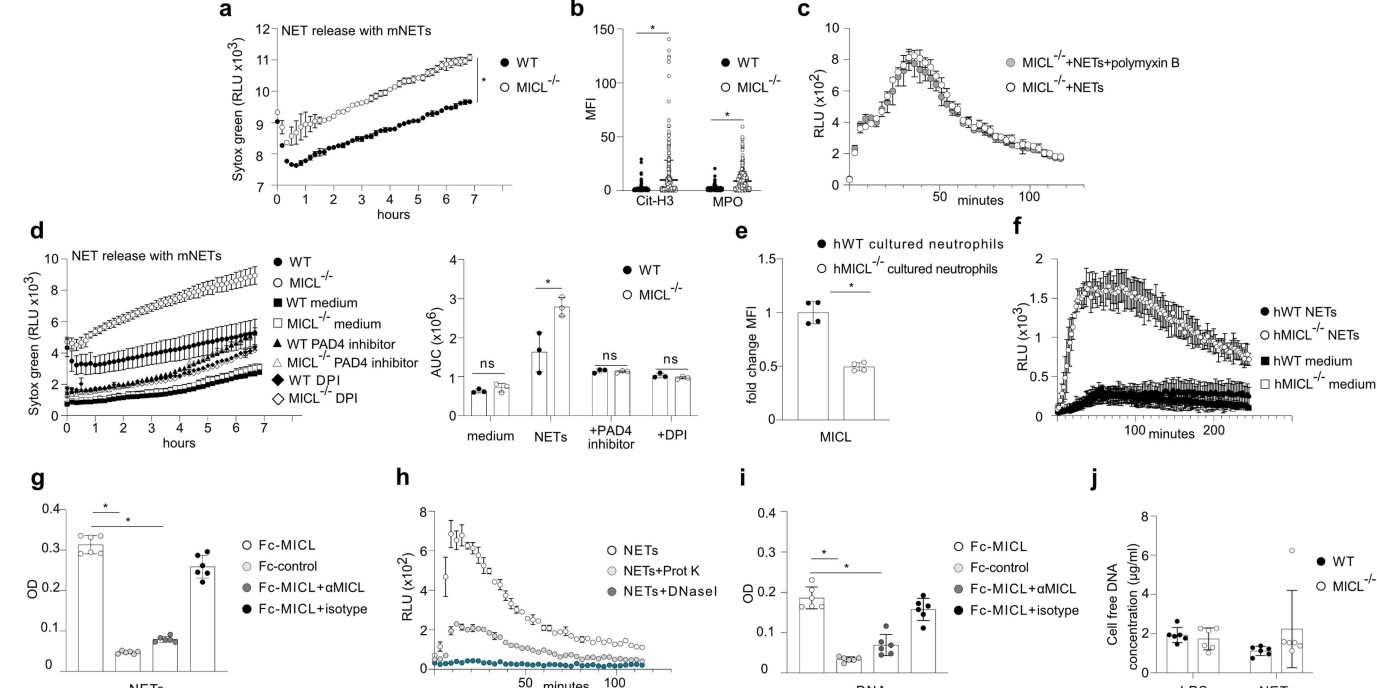

**Extended Data Fig. 7 | MICL recognizes NETs. a**, NET-bound Sytox green fluorescence of bone marrow-isolated neutrophils stimulated with preformed murine NETs (mNETs). Data is a representative example of n = 3 independent experiments, mean ± SD performed in triplicate. **b**, Mean fluorescence intensity (MFI) of cit-H3 and MPO (mean ± SD, 3 fields of view per condition) during NET formation in WT or MICL-deficient mNETs-stimulated neutrophils. Statistical significance determined by Student's t-test. **c**, ROS production of MICL-deficient neutrophils mNETs-stimulated with or without polyxymin B. Data is a representative example of n = 2 independent experiments, mean ± SD performed in triplicate. **d**, NET formation and quantification of area under curve (AUC, right) of WT or MICL-deficient mNETs-stimulated neutrophils with GSK484 or DPI. Data is a representative example of n = 2 independent experiments, mean ± SD performed in triplicate. **e**, Fold change in MFI of MICL expression on cultured neutrophils (CD66b⁺CD15⁺), relative to human neutrophils treated with negative control scrambled guide RNA. **f**, ROS production by preformed NETs of human MICL-knockout cultured neutrophils

(hKO) or human WT derived from CD34⁺ haematopoietic progenitors. Data is a representative example of n = 2 independent experiments, mean ± SD performed in triplicate. **g**, Fc-MICL NET recognition by ELISA in the presence/absence of anti-MICL antibodies. Pooled data represented as mean ± SD of n = 2 independent experiments performed in triplicate. Statistical significance determined by One-way ANOVA and Bonferroni multiple comparison test. **h**, ROS production of MICL-deficient neutrophils stimulated with untreated NETs, Proteinase K-treated NETs or DNaseI-treated NETs. Data is a representative example of n = 3 independent experiments, mean ± SD performed in triplicate. **i**, Fc-MICL genomic DNA recognition by ELISA in the presence/absence of anti-MICL antibodies. Pooled data represented as mean ± SD of n = 2 independent experiments performed in triplicate. Statistical significance determined by One-way ANOVA and Bonferroni multiple comparison test. **j**, Cell free DNA concentration 4 h after NETs or LPS intraperioteneal injection in WT and MICL-deficient animals (n = 1 experiment with 6 animals/group). *, p < 0.05.

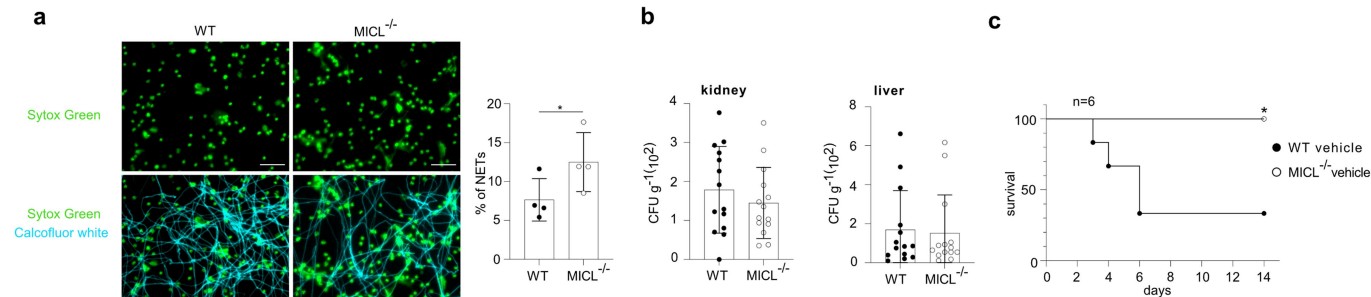

**Extended Data Fig. 8 | MICL regulates NET formation during fungal infection. a**, Representative Sytox green fluorescent (DNA, green) and calcofluor white (blue) images of *A. fumigatus*-induced NETs in bone marrow neutrophils isolated from WT or MICL-deficient animals and quantification as percentage of neutrophils under NETosis (3 fields of view per condition) at 4 h post-stimulation. Statistical significance determined by student's t-test.

**b**, Tissue fungal burdens of mice two days after intravenous (i.v) infection with $10^6$ *A. fumigatus* conidia (values shown are mean± SEM of pooled data from two independent experiments, $n$ = 14 biologically independent mice). **c**, Survival of mice following i.v infection with $10^6$ *A. fumigatus* conidia and treated with vehicle control (n = 1 experiment with 6 mice/group) analysed by log-rank test. $p$ = 0.0195. CFU, colony-forming unit. *, p < 0.05.

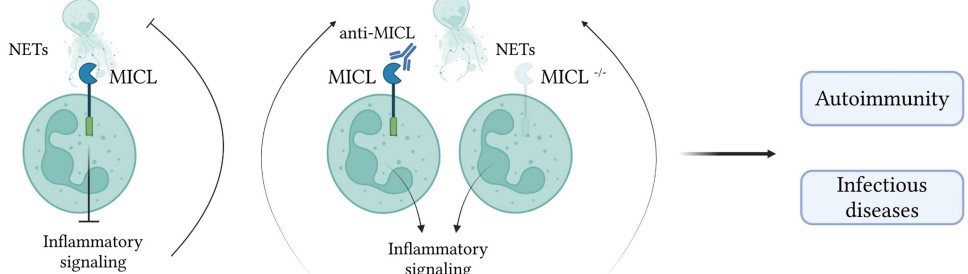

**Extended Data Fig. 9 | Proposal model for MICL in NETs recognition.** MICL directly recognizes NETs and this interaction regulates neutrophil activation. In the absence of MICL or in the presence of antibodies targeting this receptor, this interaction fails to occur, generating a positive feedback loop of neutrophil activation that on the one hand increases the severity of autoimmunity, but on the other, increases the ability to resist invasive infections. Diagram was created using BioRender (https://biorender.com).

**Extended Data Table 1 | Patient characteristics of the SERA cohort**

| SERA Cohort | |
|---|---|
| Number | 200 |
| Female, N (%) | 123 (61.5%) |
| Males, N (%) | 77 (38.5%) |
| Age years, mean (SD) | 58 ±13 |
| **Disease Parameters** | **Baseline** |
| RF positivity, N (%) | 130 (65%) |
| RF titre, median (IQR) | 72.6 (4-230) |
| CCP positivity, N (%) | 173 (86.5%) |
| αCCP titre, median (IQR) | 179 (4-340) |

# Reporting Summary

## Statistics

For all statistical analyses, confirm that the following items are present in the figure legend, table legend, main text, or Methods section.

| n/a | Confirmed | |
|---|---|---|
| ☐ | ☒ | The exact sample size (*n*) for each experimental group/condition, given as a discrete number and unit of measurement |
| ☐ | ☒ | A statement on whether measurements were taken from distinct samples or whether the same sample was measured repeatedly |
| ☐ | ☒ | The statistical test(s) used AND whether they are one- or two-sided *Only common tests should be described solely by name; describe more complex techniques in the Methods section.* |
| ☐ | ☒ | A description of all covariates tested |
| ☐ | ☒ | A description of any assumptions or corrections, such as tests of normality and adjustment for multiple comparisons |
| ☐ | ☒ | A full description of the statistical parameters including central tendency (e.g. means) or other basic estimates (e.g. regression coefficient) AND variation (e.g. standard deviation) or associated estimates of uncertainty (e.g. confidence intervals) |
| ☒ | ☐ | For null hypothesis testing, the test statistic (e.g. *F*, *t*, *r*) with confidence intervals, effect sizes, degrees of freedom and *P* value noted *Give P values as exact values whenever suitable.* |
| ☒ | ☐ | For Bayesian analysis, information on the choice of priors and Markov chain Monte Carlo settings |
| ☒ | ☐ | For hierarchical and complex designs, identification of the appropriate level for tests and full reporting of outcomes |
| ☐ | ☒ | Estimates of effect sizes (e.g. Cohen's *d*, Pearson's *r*), indicating how they were calculated |

*Our web collection on statistics for biologists contains articles on many of the points above.*

## Software and code

Policy information about availability of computer code

| | |
|---|---|
| Data collection | For Flow cytometry data collection Becton Dickinson FACSDiva v.8.0.3, SpectroFlo v.3.2.1 or INSPIRE software was used. |
| Data analysis | GraphPad Prism v.10.2.0 was used for statistical analysis<br>FlowJo v.10 was used for flow cytometry data analysis<br>Fiji was used for microscopy image analysis<br>QuPath v.0.4.3 was used for segmentation of fluorescent microscopy images |

For manuscripts utilizing custom algorithms or software that are central to the research but not yet described in published literature, software must be made available to editors and reviewers. We strongly encourage code deposition in a community repository (e.g. GitHub). See the Nature Portfolio guidelines for submitting code & software for further information.

## Data

Policy information about availability of data

All manuscripts must include a data availability statement. This statement should provide the following information, where applicable:
- Accession codes, unique identifiers, or web links for publicly available datasets
- A description of any restrictions on data availability
- For clinical datasets or third party data, please ensure that the statement adheres to our policy

All data necessary for the conclusions of this study are available in the main text, figures and Extended data. Source data are provided with the Article. Additional RA

# Research involving human participants, their data, or biological material

Policy information about studies with underline{human participants or human data}. See also policy information about underline{sex, gender (identity/presentation), and sexual orientation} and underline{race, ethnicity and racism}.

| | |
|---|---|
| Reporting on sex and gender | Based on biological attributes provided about the cohort from the Scottish Early Rheumatoid Arthritis (SERA) from the 200 samples used, 123 (61.5%) were females and 77 (38.5%) were males. |
| Reporting on race, ethnicity, or other socially relevant groupings | None of these characteristics were included in the data collected from participants of the SERA cohort (Dale, J., Paterson, C., Tierney, A. et al. The Scottish Early Rheumatoid Arthritis (SERA) Study: an inception cohort and biobank. BMC Musculoskelet Disord 17, 461 (2016))<br>None of these characteristics were included in the data collected from participants of the Covid-19 cohorts (Kusakabe T, Lin WY, Cheong JG, Singh G, Ravishankar A, Yeung ST, Mesko M, DeCelie MB, Carriche G, Zhao Z, Rand S, Doron I, Putzel GG, Worgall S, Cushing M, Westblade L, Inghirami G, Parkhurst CN, Guo CJ, Schotsaert M, García-Sastre A, Josefowicz SZ, Salvatore M, Iliev ID. Fungal microbiota sustains lasting immune activation of neutrophils and their progenitors in severe COVID-19. Nat Immunol. 2023 Nov;24(11):1879-1889. doi: 10.1038/s41590-023-01637-4) |
| Population characteristics | From the 200 samples from the Scottish Early Rheumatoid Arthritis (SERA) Study used, patients were 58 ±13 years old, 123 (61.5%) were females and 77 (38.5%) were males.<br>For the Covid-19 cohort, participants were recruited from the inpatient division of New York-Presbyterian Hospital: 25 moderate COVID-19 patients (10 males and 15 females, median age of 61.7 years), 66 severe COVID patients-19 (45 males and 21 females, median age of 66.6 years) and 36 healthy controls (16 males and 20 females, median age of 54.5 years).<br>For the SLE cohort, samples were obtained from the Imperial College Healthcare Tissue Bank (ICHTB). 40 SLE patients (40 females, median age 42.5 years) and 27 healthy controls (27 females, median age 36 years) |
| Recruitment | "Rheumatology units from across Scotland participate in the SERA study. Patients with a new clinical diagnosis of RA or UA, and who have at least one swollen joint, are invited to participate. Patients are excluded if their joint swelling can be explained by an alternative diagnosis (e.g. psoriatic arthritis) or if they are carriers of blood borne viruses. Duration of symptoms up until diagnosis is not an exclusion criterion. Potential participants are referred to local SERA research nurses for screening and baseline assessments. Treatment decisions (including initiation and escalation) and clinical follow-up remain the responsibility of the local rheumatology department who follow standard local practice. Patients are not excluded if treatment with steroids or DMARDs has already started prior to recruitment (for example by the General Practitioner) as long as the diagnosis of UA/RA is new, and treatment has commenced within the last 6 months. Participants are asked to invite a first degree relative, or friend of the same gender and similar age, to participate in a cohort of healthy controls with a similar genetic or demographic background. All participants provide generic and enduring consent that allows: collection of demographic and outcome data; retrieval and linkage of routine health care data; and long term storage of data and samples for future research projects."(Dale, J., Paterson, C., Tierney, A. et al. The Scottish Early Rheumatoid Arthritis (SERA) Study: an inception cohort and biobank. BMC Musculoskelet Disord 17, 461 (2016))<br>"Recruitment of Covid-19 samples: Participants were recruited from patients hospitalized at New York Presbyterian Hospital from March to June 2020. Some subjects were followed after recovery from mCOVID-19 or sCOVID-19 and were partitioned into an early convalescent group (2-4 months following admission) and a late convalescent group (4-12 months following admission). All patients were classified in mCOVID-19 and sCOVID-19 according with oxygen requirements with mCOVID-19 defined as SARSCoV-2 infection and <6 liters noninvasive supplemental oxygen to maintain SpO2 >92%, and sCOVID-19 defined as SARS-CoV-2 infection requiring hospitalization and received >6 liters supplemental oxygen or mechanical ventilation (Kusakabe T, Lin WY, Cheong JG, Singh G, Ravishankar A, Yeung ST, Mesko M, DeCelie MB, Carriche G, Zhao Z, Rand S, Doron I, Putzel GG, Worgall S, Cushing M, Westblade L, Inghirami G, Parkhurst CN, Guo CJ, Schotsaert M, García-Sastre A, Josefowicz SZ, Salvatore M, Iliev ID. Fungal microbiota sustains lasting immune activation of neutrophils and their progenitors in severe COVID-19. Nat Immunol. 2023 Nov;24(11):1879-1889. doi: 10.1038/s41590-023-01637-4)<br>All patients with SLE in the study met the revised American College of Rheumatology criteria [Tan EM, et al. The 1982 revised criteria for the classification of systemic lupus erythematosus. Arthritis Rheum. 25, 1271-1277 (1982)] and the SLICC [Petri M, et al. Derivation and validation of the Systemic Lupus International Collaborating Clinics classification criteria for systemic lupus erythematosus. Arthritis Rheum. 64, 2677-2686 (2012)] criteria. Some patient had a history of biopsy-proven nephritis according to the International Society of Nephrology/Renal Pathology Society classification. Healthy female volunteers (with no family history of autoimmune disease) served as age-matched and ethnicity-matched controls. |
| Ethics oversight | "The SERA Study was initiated by the Scottish Collaborative Arthritis Research (SCAR, www.scarnetwork.org) network and represents a collaboration between the Universities of Aberdeen, Dundee, Edinburgh and Glasgow, NHS Scotland, Healthcare Improvement Scotland, the Chief Scientist's Office Scotland and Pfizer Ltd. The study's protocol and procedures were reviewed and given favourable opinion by the West of Scotland Research Ethics Committee and all included patients provided written, enduring consent to participate. The study is managed by a scientific steering committee comprising clinicians and academics, from each of the participating NHS Health Boards and Universities, and (until April 2015) representatives of Pfizer Ltd." (Dale, J., Paterson, C., Tierney, A. et al. The Scottish Early Rheumatoid Arthritis (SERA) Study: an inception cohort and biobank. BMC Musculoskelet Disord 17, 461 (2016))<br>-Serum samples were also obtained from consenting healthy donors with the approval of the College of Life Sciences and Medicine ethics review board, University of Aberdeen (Application number 1243)<br>"Covid-19 patients research was reviewed and approved by the Institutional Review Board of Weill-Cornell Medicine (New York Presbyterian and Lower Manhattan hospitals) (#IRB 20-03021645 and #IRB 20-03021671). Informed consents were obtained from all enrolled patients and healthcare workers by trained staff and records maintained in research database for the duration of our study" (Kusakabe T, Lin WY, Cheong JG, Singh G, Ravishankar A, Yeung ST, Mesko M, DeCelie MB, Carriche |

Note that full information on the approval of the study protocol must also be provided in the manuscript.

# Field-specific reporting

Please select the one below that is the best fit for your research. If you are not sure, read the appropriate sections before making your selection.

☒ Life sciences ☐ Behavioural & social sciences ☐ Ecological, evolutionary & environmental sciences

For a reference copy of the document with all sections, see nature.com/documents/nr-reporting-summary-flat.pdf

# Life sciences study design

All studies must disclose on these points even when the disclosure is negative.

| | |
|---|---|
| Sample size | Sample sizes of at least five per group were chosen as this would allow the detection of a 25% difference in the mean between experimental and control groups with a probability of greater than 95% (p < 0.05), assuming a standard deviation of around 15% and a minimum power value of 0.8. |
| Data exclusions | Extended Data Fig 1a. One mouse was excluded from analysis due to lack of response<br>Extended data Fig 6c. 3 animals were excluded since they were euthanized during the experiment |
| Replication | All experiments were independently replicated at least once unless otherwise indicated in the manuscript. |
| Randomization | Mouse experiments were performed by random assignation of age- and sex-matched mice in experimental groups at the beginning of each experiment to experimental or control groups, females were co-housed, and experiments were not blinded.<br>For in vitro experiments, samples were randomly assigned to experimental or control groups. |
| Blinding | Experiments were not blinded, as the investigator who planned the experiments, also performed them but were conducted according to standardized protocols and procedures. |

# Reporting for specific materials, systems and methods

We require information from authors about some types of materials, experimental systems and methods used in many studies. Here, indicate whether each material, system or method listed is relevant to your study. If you are not sure if a list item applies to your research, read the appropriate section before selecting a response.

## Materials & experimental systems

| n/a | Involved in the study |
|---|---|
| ☐ | ☒ Antibodies |
| ☐ | ☒ Eukaryotic cell lines |
| ☒ | ☐ Palaeontology and archaeology |
| ☐ | ☒ Animals and other organisms |
| ☒ | ☐ Clinical data |
| ☒ | ☐ Dual use research of concern |
| ☒ | ☐ Plants |

## Methods

| n/a | Involved in the study |
|---|---|
| ☒ | ☐ ChIP-seq |
| ☐ | ☒ Flow cytometry |
| ☒ | ☐ MRI-based neuroimaging |

# Antibodies

| | |
|---|---|
| Antibodies used | anti-CD45-FITC (Clone 102), anti-CD45-PerCP-Cyanine5.5(Clone 102), anti-CD11b-BUV395(Clone M1/70), anti-CD11b-PE-Cy7 (Clone M1/70), anti-GR-1-APC (Clone RB6-8C5), anti-MHC-II-FITC (Clone 2G9), anti-MHC-II-BUV496 (Clone 2G9)anti-C5aR-BV510 (Clone 20/70), anti-Ly6G-BV421(Clone 1A8), anti-Ly6G-Spark Blue 550(Clone 1A8), anti-Ly6GAPC (Clone 1A8), anti-CD62L-BV510 (Clone MEL-14), anti-CD18-BV650 (Clone C71/16), anti-CD18-APC (H155-78), anti-F4/80-AF700 (Clone BM8), anti-F4/80-PECy7 (Clone BM8), anti-Ly6C-PE-Cy7 (Clone HK1.4), anti-Ly6C-Brilliant Violet 570 (Clone HK1.4), anti-CCR1-PE (Clone 643854), anti-CXCR2-APC (Clone SA045E1), anti-CD11c-BV711 (Clone HL3), anti-B220-AlexaFluor700 (Clone RA3-6B2), isotype control AFRC MAC 49 (ECACC 85060404; isotype for anti-MICL), anti-CD66-PE/Dazzle (Clone G10F5), anti-CD15-AF700 (Clone Hi98), anti-CD16-APC (Clone 3G8) and HRP-conjugated goat F(ab')2 fragment anti-human IgG (JacksonImmuno Research). Remaining antibodies were purchased commercially from eBioscience, R&D systems or Biolegend. |

anti-mMICL (309), isotype control AFRC MAC 49 (ECACC 85060404; isotype for anti-mMICL), anti-hMICL (HB3) and isotype control D1.3 (isotype for anti-hMICL) were generated in house.
anti-citrullinated histone 3 (ab5103, Abcam), anti-Myeloperoxidase (AF3667, R&D) and anti-DNA/H1 (AB3864, Merck) were used for Immunofluorescence staining of NETs
To achieve neutrophil depletion rat anti-mouse Ly6G (Clone 1A8), rat anti-mouse GR-1 (Clone RB6-8C5) or isotype controls (rat anti-mouse IgG2a, rat anti-mouse IgG2b). All were purchased commercially from Bio-X-cell.

| Validation | Anti-mMICL antibodies were generated and validated as follows. Sprague Dawley rats were immunised with Fc-mMICL in Freund's complete adjuvant. After a final intraperitoneal boost, without adjuvant, rat splenocytes were harvested and fused with Y3 myeloma cells, as described. Hybridoma supernatants were screened by ELISA and positives were then tested by immunohistochemistry and flow cytometry, as described below, against Fc-mMICL as well as mMICL transduced NIH3T3 fibroblasts. |

The mAb, HB3, specific for hMICL, was generated by immunization of C57BL/6 mice with an Fc-hMICL fusion protein. Hybridomas were generated according to standard protocols and supernatants from clonally diluted cells were screened by ELISA. The mAb HB3 (IgG1) was subsequently selected based on its ability to function in FACS, Western blot, and immunocytochemistry.
Remaining antibodies were well validated commercial clones and routinely QC'ed by the manufacturer.Please refer to the spec sheets on the respective vendors' website for technical information and detail by searching the catalog numbers or clone numbers provided above.

## Eukaryotic cell lines

Policy information about cell lines and Sex and Gender in Research

| Cell line source(s) | HEK293T, NIH3T3 cell lines were originally purchased from the ATCC. BWZ.36 NFAT-lacZ were kindly provided by Wayne Yokoyama, Washington, USA |
| Authentication | No authentication methods were used. |
| Mycoplasma contamination | Mycoplasma tests were done historically on these cell lines, but not for the experiments detailed in the manuscript. |
| Commonly misidentified lines (See ICLAC register) | none are listed as misidentified |

## Animals and other research organisms

Policy information about studies involving animals; ARRIVE guidelines recommended for reporting animal research, and Sex and Gender in Research

| Laboratory animals | C57BL/6 and Clec12a -/- mice (6-8 weeks old) were obtained from the specific pathogen-free facility at the University of Aberdeen or Charles River Laboratories. Animal experiments were performed using age-matched female or male mice and conformed to the animal care and welfare protocols approved by UK Home Office (Project license numbers: P79B6F297 and P6A6F95B5) in compliance with all relevant local ethical regulations. Clec12a-/- mice were generated commercially (Taconic Artemis) by conventional gene targeting in C57BL/6 embryonic stem cells (Redelinghuys P, Whitehead L, Augello A, et al. MICL controls inflammation in rheumatoid arthritis, Annals of the Rheumatic Diseases 2016;75:1386-1391). Male DBA/1OlaHsd mice (6-8) weeks old were obtained from Inotiv and maintained at the University of Exeter. |
| Wild animals | The study did not involve wild animals |
| Reporting on sex | Collagen type II antibody-induced arthritis (CAIA) and Collagen-induced arthritis (CIA) were originally described in males as they have have enhanced susceptibility compared to females (Holmdahl R, Jansson L, Larsson E, Rubin K, Klareskog L: Homologous type II collagen induces chronic and progressive arthritis in mice. Arthritis Rheum 1986, 29:106-113; Nandakumar KS, Svensson L, Holmdahl R. Collagen type II-specific monoclonal antibody-induced arthritis in mice: description of the disease and the influence of age, sex, and genes. Am J Pathol. 2003). Data in this paper shows CAIA, CIA and K/BxN serum transfer models performed in males. CAIA data was also confirmed in females. |
| Field-collected samples | The study did not involve samples collected from the field |
| Ethics oversight | All experiments conformed to the ethical review committee of the University of Aberdeen, University of Exeter, and the UK Home Office regulations (Project license numbers: P79B6F297 and P6A6F95B5). |

Note that full information on the approval of the study protocol must also be provided in the manuscript.

# Flow Cytometry

## Plots

Confirm that:

☒ The axis labels state the marker and fluorochrome used (e.g. CD4-FITC).

☒ The axis scales are clearly visible. Include numbers along axes only for bottom left plot of group (a 'group' is an analysis of identical markers).

☒ All plots are contour plots with outliers or pseudocolor plots.

☒ A numerical value for number of cells or percentage (with statistics) is provided.

## Methodology

| | |
|---|---|
| Sample preparation | Murine cells and tissues:<br>Murine peripheral blood leukocytes, thioglycolate-elicited inflammatory peritoneal cells and bone marrow cells were isolated essentially as described previously (Taylor PR et al Eur J Immunol, 2003).<br>Bone marrow neutrophils were isolated using a gradient of Histopaque separation media (Merck) by a density gradient centrifugation method or using the EasySep™ Mouse Neutrophil Enrichment Kit (STEMCELL Technologies).<br>Cells were isolated from the hind paw ankle joint of arthritic mice using the protocol described by Armaka et al 2009. The isolated tissue was incubated 60 minutes with Collagenase VIII (Sigma-Aldrich). Cells were strained through 70 μm nylon cell strainers (Fisher Scientific) and collected by centrifugation.<br><br>NNIH3T3 fibroblasts stably expressing full length murine or human MICL were maintained at 37°C and 5% CO2 in DMEM or RPMI medium supplemented with 10% heat-inactivated foetal calf serum, 100 units/mL penicillin, 0.1 mg/mL streptomycin, and 2 mM L-glutamine.<br><br>For flow cytometry, single cell suspensions were stained with fixable viability dye efluor 780 (eBioscience), and washed in FACS wash (PBS with 0.5% (w/v) BSA and 5-10 mM EDTA) containing anti-CD16/CD32 (Clone 2.4G2, prepared in house). The following antibodies were used for FACS analysis following standard methodology: anti-CD45-FITC (Clone 102), anti-CD45-PerCP-Cyanine5.5(Clone 102), anti-CD11b-BUV395(Clone M1/70), anti-CD11b-PE-Cy7 (Clone M1/70), anti-GR-1-APC (Clone RB6-8C5), anti-MHC-II-FITC (Clone 2G9), anti-MHC-II-BUV496 (Clone 2G9), anti-C5aR-BV510 (Clone 20/70), anti-Ly6G-BV421(Clone 1A8), anti-Ly6G-Spark Blue 550(Clone 1A8), anti-Ly6GAPC (Clone 1A8), anti-CD62L-BV510 (Clone MEL-14), anti-CD18-BV650 (Clone C71/16), anti-CD18-APC (H155-78), anti-F4/80-AF700 (Clone BM8), anti-F4/80-PECy7 (Clone BM8), anti-Ly6C-PE-Cy7 (Clone HK1.4), anti-Ly6C-Brilliant Violet 570 (Clone HK1.4), anti-CCR1-PE (Clone 643854), anti-CXCR2-APC (Clone SA045E1), anti-CD11c-BV711 (Clone HL3), anti-B220-AlexaFluor700 (Clone RA3-6B2), and anti-CD3-Alexa Fluor 647 (Clone 17A2) a. All were purchased commercially from eBioscience, R&D systems or Biolegend. αmClec12a-biotinylated, and isotype control AFRC MAC 49 (ECACC 85060404; isotype for anti-mMICL) was generated in house.<br><br>Human neutrophils from blood of healthy donors were purified using a Ficoll-Paque density centrifugation method (Kuhns DB et al, Curr. Protoc. Immunol. 2015) or using the EasySepTM direct human neutrophil isolation kit (STEMCELL Technologies) as per the manufacturer's instructions. Single cell suspensions were stained with fixable viability dye Zombie Aqua. Cells were washed in FACS wash (PBS with 0.5% (w/v) BSA and 5-10 mM EDTA) and stained with anti-CD66-PE/Dazzle (Clone G10F5), anti-CD15-AF700 (Clone Hi98), anti-CD16-APC (Clone 3G8) and anti-hMICL or isotype control antibodies. All were purchased commercially from Biolegend. αhMICL was generated in house. |
| Instrument | BD LSR II Fortessa flow cytometer (BD Biosciences), Cytek Aurora Spectral cytometer (Cytek), Amnis Image StreamX imaging flow cytometer |
| Software | For collection Becton Dickinson FACSDiva, SpectroFlo software and INSPIRE software were used. For analysis FlowJo software was used. |
| Cell population abundance | No cell populations were sorted for this manuscript. |
| Gating strategy | Gating strategy for murine tissue staining<br>Using the FSC/SSC plot, a gate was drawn to select cells and exclude debris.<br>Using the FSC-H/FSC-A plot, a gate was drawn to select single cells.<br>Using the FSC-H/Efluor780 plot, a gate was drawn to select viable cells.<br>Using the FSC-H/ PerCP-Cy5.5 plot, we gated on CD45 negative and positive cells.<br>Using the Spark Blue 550/BUV395 plot, we gated on Ly6G CD11b double positive cells, CD11b positive Ly6G negative cells and Ly6G negative CD11b negative cells.<br>From the D11b positive Ly6G negative cells, using the PECy7/ BUV395 we gated on F4/80 positive cells.<br>From the F4/80 positive cells, using the Brilliant Violet 570 /BUV395 we gated on Ly6C high and Ly6C low cells.<br>From the Ly6G negative CD11b negative cells, using the BV605/Alexa Fluor 647 we gated on B220 positive or CD3 positive cells.<br>Positive and negative populations for gating were defined using FMO and isotype controls.<br><br>Gating strategy for cultured neutrophils<br>Using the FSC/SSC plot, a gate was drawn to select cells and exclude debris.<br>Using the FSC-H/FSC-A plot, a gate was drawn to select single cells.<br>Using the FSC-H/BV510 plot, a gate was drawn to select viable cells. |

Using the PE-Dazzle/Alexa Fluor 700 plot we gated on CD66b CD15 double positive cells. We used this gate to detect MICL expression using the Alexa Fuor 488 channel.

☒ Tick this box to confirm that a figure exemplifying the gating strategy is provided in the Supplementary Information.

