## [Peer Review File · Nature]

Manuscript Title: Recognition and control of Neutrophil Extracellular Trap formation by MICL

Reviewer Comments & Author Rebuttals

Reviewer Reports on the Initial Version:

Referees' comments:

Referee #1 (Remarks to the Author):

This manuscript suggests that MICL downregulates NETosis. While some work has already been published on other receptors that suppress NETosis, such as Siglecs, the authors propose that in contrast to these receptors that interact with microbes, MICL recognises NETs directly and acts as a NET sensor. In this way the authors diverge from previously published work that implicates MICL in direct MSU recognition.

However, the model is not only based on the notion that MICL blocks NETosis in response to exogenous signals such as MSU crystals. Instead, the authors implicate MICL in the downregulation of NETosis in response to stimulation by NETs which is a novel concept. This is an important divergence from published literature (MSU/MICL Immunity 2014 paper), which would otherwise negatively impact on the novelty of the findings in this manuscript.

Moreover, known positive feedback loops in NETosis involved primarily indirect amplification of inflammation via the proinflammatory capacity of NETs. But a direct role of NETs in triggering further NETosis had not been previously explored and raises important questions for the requirement of a regulatory receptor that would limit such pathogenic feedback loops.

Overall, this is an interesting hypothesis with important implications that needs to be more thoroughly explained in the manuscript. For instance, the data relating to NETosis stimulation by NETs are not stated in the abstract. There are important mechanistic implications that stem from this that need to be explored, since this dual role of NETs complicates the story, because in some experiments NETs are both a trigger and a suppressor.

The relevance for human disease driving mechanisms stemming from the analysis of anti-MICL antibodies in Figure 4 is also very intriguing and would further elevate the abstract and the novelty and significance of the findings as well.

On the technical side, the role of MICL in regulating NETosis in vitro and in vivo requires more robust data as important representative microscopy is not very convincing and lacks quantification. Moreover, the in vitro NETosis quantification assay employed here does not distinguish between NETosis and necrosis, and microscopy-based quantification should be employed instead. This leads to gross discrepancies between the NET-stimulated ROS burst and so called NETosis (Sytox fluorescence) results that needs to be addressed.

Lastly, the in vivo fungal infection experiments are provocative, but need further supplementation.

Specific remarks:

1) Overall, the in vivo data on the role of MICL in murine arthritis in Figures 1 and 2 are convincing, but most of it has already been previously reported. The link with neutrophils is convincing, but that was expected. I think the adoptive transfer in Figure 2 is the critical experiment in these figures given the absence of a conditional neutrophil specific KO in the study.

2) Figure 3. This is where the interesting results begin. In Fig. 3A it is striking that the KO has not impact on PMA. This suggests that the regulation of neutrophil activation by MICL acts upstream of ROS and is specific to different stimuli.

The zymosan experiment is very important here because up to this point, the data are consistent with the role of MICL as a MSU crystal receptor. This is the first piece of evidence that the receptor also suppresses ROS “allosterically” in response to other non-crystal signals. For this reason, the authors should try hyphae instead of zymosan, since hyphae induce a ROS burst that is extracellular and much more relevant for NETosis (PMID: 28314592, 25217981).

3) Fig. 3B. The difference in NETosis is not very clear from these representative images. The authors need to also stain with cell permeable dyes to visualise the total number of neutrophils (Hoechst, DAPI...). The images should not be fixed but rather imaged undisturbed, to avoid NET stretching and aggregating which distorts them and renders NET quantification nearly impossible. NETs should appear as individual clouds around the neutrophil corpses interacting with crystals that can then be easily quantified.

4) Figure 3C. I do not detect a great deal of difference between the number of cit-H3 signal events in the WT and KO. Individual channels should be presented and co-stained with Ly6G that also stains NETs, but enumerates neutrophils. Slide scanner image processing quantifications of cit-H3/Ly6G area or event ratios are the best way to objectively quantify NETs in vivo, accompanied by cit-H3 and MPO WB of tissue or synovial lysates. If the ABCAM antibody is employed, it is also important that a basic rather than neutral antigen retrieval protocol is implemented to avoid background artifacts with this antibody. Scale bars are also missing.

5) Figure 3D. The authors should provide raw images from the imaging flow cytometry quantification of NETs. This is the first time I encounter this method of NET quantification and given how fragile NETs are, it is difficult to conceive how this technique can work reliably, particularly for in vivo sample analysis, without driving NET shearing and aggregation with large bundles of live cells. Quantification of immunofluorescence microscopy tissue images stained for DNA/H1 and Ly6G stains will be more reliable in my opinion (as in 3C). It is very likely that PAD4 inhibition will not block NET formation (only their citrullination) but can still inform on the impact of NET-derived histones whose activity is amplified by citrullination.

6) Fig. 3G. This is a nice result. However, citrullination of other antigens in a NET-independent manner has also been implicated in RA pathogenesis (PMID: 36658110). It is conceivable that MICL

blocks this alternative mechanism as well. A second validation with DNase I treatment instead of PAD4 inhibition would thus be helpful. Histone blocking might also be effective, although more expensive. NE inhibitors are a third option. Which PAD4 inhibitor was used in these experiments (3D-G)? Comparing PAD4 to PAD2 specific inhibitors in the RA model may also help show that the phenotype is linked to NET-related citrullination which is PAD4 dependent.

7) Figure 4B and 5B. This assay only reports on dead neutrophils rather than specifically on NETosis. Microscopy-based chromatin decondensation measurements are the most accurate approach.

8) Figure 5A, 5C. In addition to DNase I treated controls (Supplement), polymyxin B treated controls should be included to ensure that the NET preparations used for stimulation of ROS are working via NETs rather than via LPS contamination.

9) Supp. Fig. 10B and 5B. In this experiment the MICL KO and WT neutrophils activation curves are nearly identical. The way this is presented at first it seems to contradict the result in fig. 5B. Looking back at the RFU scale in Figure 5B, it looks as if the actual differences in signal are approximately 10% at any timepoint in both graphs. It also appears that the curve in 5B is zoomed in and that the actual starting time on the x axis is not 0 as is the case in Supp. 10B, or that there is a delay in reading this experiment. The entire curve and full axis in 5B should be shown. In any event, this assay needs to be supplemented with direct NET quantification by microscopy.

10) To accompany the PAD4 and DPI inhibition experiments, the authors should also provide microscopy images of NET-triggered NETs stained for NE, MPO, DNA and cit-H3 antibodies by microscopy and WB, to show that NETs trigger chromatin citrullination.

11) Sup. Fig. 10A and 5A. Unstimulated controls should be shown to ascertain the degree to which WT neutrophils are stimulated to produce ROS by NETs. In Fig. S10A it appears that WT neutrophils do not produce any ROS, but in Fig. 5A there is some ROS production by WT cells. Adding unstimulated cells will help ascertain whether WT neutrophils respond at all to NETs in terms of ROS production, or whether this is only observed exclusively in KO cells.

12) Fig.5 and S10. Related to the the previous point, there is a large discrepancy between the ROS production and NETosis assays. NETs barely activate ROS production in WT cells, but the difference in NET formation (by Sytox bulk fluorescence) is very small (10%). Thus, if WT neutrophils barely respond to NETs ROS-wise, and DPI blocks the process (Fig. S10B) why are the WT cells making 90% of the NETs that a KO neutrophil is making in response to NETs? I suspect that this is further evidence that the Sytox fluorescence assay is not the proper approach here.

13) The lack of Proteinase K effect suggests that the interaction depends on DNA but not protein (Fig. 5D). Thus, MICL could be a DNA sensor and this could be tested with genomic neutrophil DNA alone or perhaps DNA oxidised in vitro. Moreover, genomic DNA does not trigger NETosis, but ROS stimulation is dependent on both DNA and protein (Fig. S10C). Falling short of dissecting the molecular determinants of NET-mediated stimulation, the authors could take advantage of these differences to further dissect the two activities. If MICL binds naked DNA, then it would be

interesting to explore whether MSU crystal-triggered NETosis is blocked by naked DNA titrations. Naked DNA could also block NET-triggered NETosis if added in excess.

14) Neutrophil density experiments might be interesting here. Increasing neutrophil confluence inhibits NET formation in response to PMA (unpublished observations). So, one could imagine that in the MSU or even PMA stimulation experiments, MICL KO neutrophils would be less sensitive to this unexplained density suppressing phenomenon. I don't think this is an essential experiment but it would support an important concept of NET density sensing which lingers below the surface in the manuscript.

15) WB for cit-H3 and cell free DNA measurements would be helpful in the peritoneal inflammation model.

16) Fig. 6A It is strange that murine WT neutrophils do not make any "NETs" in response to *A. fumigatus*. (Again there is the issue that Sytox fluorescence readers do not distinguish between NETosis and necrosis, without microscopy data). The peak of NETosis as suggested by the fluorescence intensity in the MICL KO is consistent with what should be expected for NETosis by WT neutrophils peaking between 4-8 hrs for hyphae. This is best visualized by video timelapse, but stills at 7-8 hrs will also be useful.

17) Fig. 6B and C. These experiments should be supported with cfu (kidney and spleen), NET stains (kidney) and systemic cytokines from day 2, rather than relying on indirect inferences from PAD4 inhibitor treated mice to understand the mechanism of MICL protection. These are important readouts because the authors state that:

"we confirmed that the increased survival of MICL-deficient mice was associated with higher NET formation, since the treatment of these animals with a PAD4 inhibitor increased the susceptibility to *A. fumigatus* infection to wild type levels (Fig. 6C)." line 240

This interpretation of this experiment is oversimplified in the text because the mice in 6C should also be compared to their respective vehicle. Depending on variability across experiments, it appears that when one compares across 6B and 6C, PAD4-inh has opposing effects on WT and MICL KO mice, with WT being more protected and MICL KO being more susceptible upon PAD4 inhibition. If differences between the inhibitor and vehicle prove significant within each genotype, this would imply that the mechanisms of pathology also vary. So having the CFUs and NET stains, or cit-H3 WB of kidney lysates, and circulating un-citrullinated histones along with cytokines, IL-6, IL-1, IL-1RA (PMID:37059107) and G-CSF (PMID: 35945238) can help understand how NETosis, the cytokine storm and fungal control are affected by MICL.

18) fig. 6A The ficoll gradient employed to isolate human neutrophils is notorious for pre-stimulating the cells. This may explain why the isotype in human cells is much more reactive than WT murine neutrophils. This can be determined with unstimulated controls which are absolutely necessary here, and the human data could be vastly improved if Histopaque/Percoll is used instead.

Referee #2 (Remarks to the Author):

This study demonstrates that MICL is a regulator of NET release from neutrophils. A series of experiments show that in an arthritis model there are more neutrophils recruited and there is more injury in the MICL deficient mice. The work shows an association between NETs, tissue injury and MICL antibodies in patients. The work also suggests that NETs bind MICL and that in a fungal infection not having MICL is of benefit.

There are two main issues that would need to be completely addressed in this paper to be acceptable. First, the mechanism by which MICL leads to less NET production is missing. While the authors show lots of increased neutrophil activation and they show more NET release the mechanism by which MICL regulates this is unclear. The authors show that MICL binds the DNA, but that is where the story ends. It should start there.

The second issue is related to the first and it is that the authors have published a preliminary study showing that arthritis is worse in MICL deficient mice, that this is related to myeloid cells and that arthritis patients have increased MICL antibodies. As such without addressing the first point and finding significantly more mechanistic results the work is only incremental based on their previous work.

Specific

There are additional important issues including what does the antibody to MICL do, does it activate the MICL to reduce neutrophil activation or does it block binding of DNA. This is not well developed.

I also found it difficult to understand how the neutrophils were having similar amounts of chemokine receptors and other molecules and yet were recruited in greater numbers in MICL mice. A time course of neutrophil recruitment and NET production would address if indeed there is neutrophil recruitment that leads to NET release which recruits more neutrophils over time to lead to more NET release without MICL. The authors claim MICL regulates NET release presumably turning off this potent mechanism of action but with a single time point it is hard to understand how this conclusion can be made.

Referee #3 (Remarks to the Author):

This is an interesting paper suggesting that CLEC12A is a receptor for NETS and regulate arthritis. There are however some flaws in experimental settings and description of results.

- 1) The genetic background and experimental details of the MICL^{-/-} are not clear. Is this mouse on the same background as the used C57Bl mouse? Are they made in a B6 ES cell or with crisp? Are the experimental animals littermates? Are the experiments performed with the groups mixed in cages? Do the experiments follow the ARRIVE rules or something similar?
- 2) Its not clear whether the observed expansion and activation of PMN in the joints were the results of a direct effect by the absence of Clec12a or if it was secondary to the arthritis per se. In Fig 1 the data are based on three different experiments, is it only one selected experiment shown or are the data normalized and shown for all used mice? Is the data is selected all data need to be shown, either normalized or separate, how can we otherwise know which experiment is selected? It seems that similar problems are in all showed in vivo experiments. Apparently the increased arthritis susceptibility of the MICL mice has been previously reported – so, what is new? Its not very surprising that there are more activated PMNs in an exaggerated CAIA arthritis.
- 3) The transfer of BM PMN is novel and indicate that the Clex12a effect is restricted to PMN. However, it's not clear how pure the transferred population is. PMN are sensitive cells whereas other BM cells are not, it's difficult (but important) to exclude other BM cells such as macrophages in these experiments. If there are 10% other cells, monocytes/macrophages etc, these could explain the effect.
- 4) MSU, but not PMA induced enhanced ROS in clec12a deficient PMN. Not surprisingly it induces more NET formation. An increased NET formation was seen in arthritis joints (again not so surprising as this is expected because of more PMN in an arthritic joint). Surprisingly PAD4 inhibitors blocked the development of NETs (and arthritis) only in MICL knockout mice and not in wildtype? Why not in wildtype? Also wt has NETS.
- 5) It's not correct to state that CIA gives low penetrance in B6 mice. In fact, it is not possible to induce CIA in B6 mice. The observed arthritis in B6 mice after immunization with collagen is due to a contamination. There is a however a problem with DBA/1 as well since the authors regroup the mice after puberty, this will commonly induce a spontaneous form of arthritis or enhanced CIA, which is due to other factors than collagen autoimmunity. The experimental settings need to be different to avoid this actually well described phenomena.
- 6) The statement that antibodies to Clec12a correlates with RA is not well funded. To me it seems to be part of a known increase of natural autoantibodies in RA, since it correlates well with RF.
- 7) The definition of Clec12a as a receptor for NETS, or rather DNA within NETs is maybe not that surprising but is likely to lack specificity. Many CLRs, and other proteins, do the same. Controls for specificity is lacking.
- 8) It is well know that a major regulatory gene in autoimmune disease and RA, as well as its mouse models, is Ncf1. Mutations in Ncf1 leads to increased arthritis, decreased ROS and lack of NETs. If Clec12a is uniquely promoting arthritis through NETs it should be possible to do an experiment with Ncf1 mutated mice i.e crossing the MICL knockout with the Ncf1 mutation and investigate the littermates. Or preferably use LysG promotor conditioned MICL knockout (or conditioned knockin on MICL knockout background) mice.

Author Rebuttals to Initial Comments:

Response to Reviewers comments:

We thank the Referees for the thoughtful and constructive evaluation of our manuscript. We have highlighted all major changes in yellow in the manuscript.

Referee #1 (Remarks to the Author):

This manuscript suggests that MICL downregulates NETosis. While some work has already been published on other receptors that suppress NETosis, such as Siglecs, the authors propose that in contrast to these receptors that interact with microbes, MICL recognises NETs directly and acts as a NET sensor. In this way the authors diverge from previously published work that implicates MICL in direct MSU recognition.

However, the model is not only based on the notion that MICL blocks NETosis in response to exogenous signals such as MSU crystals. Instead, the authors implicate MICL in the downregulation of NETosis in response to stimulation by NETs which is a novel concept. This is an important divergence from published literature (MSU/MICL Immunity 2014 paper), which would otherwise negatively impact on the novelty of the findings in this manuscript. Moreover, known positive feedback loops in NETosis involved primarily indirect amplification of inflammation via the proinflammatory capacity of NETs. But a direct role of NETs in triggering further NETosis had not been previously explored and raises important questions for the requirement of a regulatory receptor that would limit such pathogenic feedback loops.

Overall, this is an interesting hypothesis with important implications that needs to be more thoroughly explained in the manuscript. For instance, the data relating to NETosis stimulation by NETs are not stated in the abstract. There are important mechanistic implications that stem from this that need to be explored, since this dual role of NETs complicates the story, because in some experiments NETs are both a trigger and a suppressor.

- We thank the reviewer for this comment and have now modified the text to better explain our hypothesis and expanded our data on underlying mechanisms of MICL function and how this is modulated by autoantibodies.

The relevance for human disease driving mechanisms stemming from the analysis of anti-MICL antibodies in Figure 4 is also very intriguing and would further elevate the abstract and the novelty and significance of the findings as well.

- We thank the reviewer for this comment and have now included these results in the abstract. Notably, we now have evidence that autoantibodies to the receptor are also present in other NET-related autoimmune diseases, including LUPUS and severe-COVID 19 patients. Moreover, we show that these antibodies functionally modify MICL function, promoting inflammatory responses.

On the technical side, the role of MICL in regulating NETosis in vitro and in vivo requires more robust data as important representative microscopy is not very convincing and lacks

quantification. Moreover, the in vitro NETosis quantification assay employed here does not distinguish between NETosis and necrosis, and microscopy-based quantification should be employed instead. This leads to gross discrepancies between the NET-stimulated ROS burst and so called NETosis (Sytox fluorescence) results that needs to be addressed.

- We addressed this relevant point using microscopy-based quantification of NET induction by NETs and the results are shown in Figure 4B and Supplementary Figure 10B. In addition, we include quantification of NET formation in the joints of arthritic mice in Supplementary Figure 7E.

Lastly, the in vivo fungal infection experiments are provocative, but need further supplementation.

- In the new version of the manuscript, we include significant additional data (fungal burdens in different tissues, in the presence and absence of PAD4 inhibitors, as well as relevant systemic and tissue cytokine levels) confirming that during invasive aspergillosis MICL restrains systemic inflammation.

Specific remarks:

1) Overall, the in vivo data on the role of MICL in murine arthritis in Figures 1 and 2 are convincing, but most of it has already been previously reported. The link with neutrophils is convincing, but that was expected. I think the adoptive transfer in Figure 2 is the critical experiment in these figures given the absence of a conditional neutrophil specific KO in the study.

- As suggested by the reviewer, we have modified Figure 1 and 2 to remove previously reported observations and include only new data (showing that MICL is required to control neutrophil responses in mouse models of arthritis).

2) Figure 3. This is where the interesting results begin. In Fig. 3A it is striking that the KO has not impact on PMA. This suggests that the regulation of neutrophil activation by MICL acts upstream of ROS and is specific to different stimuli.

The zymosan experiment is very important here because up to this point, the data are consistent with the role of MICL as a MSU crystal receptor. This is the first piece of evidence that the receptor also suppresses ROS “allosterically” in response to other non-crystal signals. For this reason, the authors should try hyphae instead of zymosan, since hyphae induce a ROS burst that is extracellular and much more relevant for NETosis (PMID: 28314592, 25217981).

- This is a helpful suggestion by the reviewer and this data is now included in Figure 2A. This shows that MICL also regulates ROS production in response to *A. fumigatus* hyphae.

3) Fig. 3B. The difference in NETosis is not very clear from these representative images. The

authors need to also stain with cell permeable dyes to visualise the total number of neutrophils (Hoechst, DAPI...). The images should not be fixed but rather imaged undisturbed, to avoid NET stretching and aggregating which distorts them and renders NET quantification nearly impossible. NETs should appear as individual clouds around the neutrophil corpses interacting with crystals that can then be easily quantified.

- We thank the reviewer for the comment. We performed this experiment using Draq5 as cell permeable dye and Sytox Green as cell impermeable dye and have now included representative images as well as quantification in Supplementary Figure 7B.

4) Figure 3C. I do not detect a great deal of difference between the number of cit-H3 signal events in the WT and KO. Individual channels should be presented and co-stained with Ly6G that also stains NETs, but enumerates neutrophils. Slide scanner image processing quantifications of cit-H3/Ly6G area or event ratios are the best way to objectively quantify NETs in vivo, accompanied by cit-H3 and MPO WB of tissue or synovial lysates. If the ABCAM antibody is employed, it is also important that a basic rather than neutral antigen retrieval protocol is implemented to avoid background artefacts with this antibody. Scale bars are also missing.

- We have now included individual channels as well as quantification of cit-H3/GR-1 area in new Supplementary Figure 7E.

5) Figure 3D. The authors should provide raw images from the imaging flow cytometry quantification of NETs. This is the first time I encounter this method of NET quantification and given how fragile NETs are, it is difficult to conceive how this technique can work reliably, particularly for in vivo sample analysis, without driving NET shearing and aggregation with large bundles of live cells. Quantification of immunofluorescence microscopy tissue images stained for DNA/H1 and Ly6G stains will be more reliable in my opinion (as in 3C). It is very likely that PAD4 inhibition will not block NET formation (only their citrullination) but can still inform on the impact of NET-derived histones whose activity is amplified by citrullination.

- We have included examples of images from the imaging flow cytometry of NET-positive cells and viable neutrophils isolated from ankle joints of arthritic mice, stained with Cit-H3, Ly6G and MPO in the Supplementary Figure 7F. Individual channels as well as the overlay are included in the figure.

6) Fig. 3G. This is a nice result. However, citrullination of other antigens in a NET-independent manner has also been implicated in RA pathogenesis (PMID: 36658110). It is conceivable that MICL blocks this alternative mechanism as well. A second validation with DNase I treatment instead of PAD4 inhibition would thus be helpful. Histone blocking might also be effective, although more expensive. NE inhibitors are a third option. Which PAD4 inhibitor was used in these experiments (3D-G)? Comparing PAD4 to PAD2 specific inhibitors in the RA model may also help show that the phenotype is linked to NET-related citrullination which is PAD4 dependent.

- Regarding the use of PAD inhibitors in the CAIA model, the manuscript includes inhibition of PAD 2/4 by BB CL-Amidine and PAD4-specific inhibition by GSK484. Treatment of MICL-deficient animals with these inhibitors during CAIA reduced the clinical score to WT levels. In addition, as requested by the reviewer, we performed DNase I treatment during CAIA (Supplementary Figure 8D) and found that treated MICL-deficient animals had a reduction in clinical score, reaching levels similar to WT mice. This confirms that NET formation determines the severity of arthritis in the absence of MICL.

7) *Figure 4B and 5B. This assay only reports on dead neutrophils rather than specifically on NETosis. Microscopy-based chromatin decondensation measurements are the most accurate approach.*

- As requested, we have now included microscopy images of NET-triggered NETs stained for cit-H3, MPO and DNA in Figure 4B as well as quantification of the MFI of cit-H3 and MPO comparing WT and MICL-deficient neutrophils stimulated with mNETs (Supplementary Figure 10B).

8) *Figure 5A, 5C. In addition to DNase I treated controls (Supplement), polymyxin B treated controls should be included to ensure that the NET preparations used for stimulation of ROS are working via NETs rather than via LPS contamination.*

- As requested, we have included polymyxin B treated NETs in Supplementary Figure 10D and show that there is no difference in the ability to induce ROS production in KO cells compared to untreated NETs.

9) *Supp. Fig. 10B and 5B. In this experiment the MICL KO and WT neutrophils activation curves are nearly identical. The way this is presented at first it seems to contradict the result in fig. 5B. Looking back at the RFU scale in Figure 5B, it looks as if the actual differences in signal are approximately 10% at any timepoint in both graphs. It also appears that the curve in 5B is zoomed in and that the actual starting time on the x axis is not 0 as is the case in Supp. 10B, or that there is a delay in reading this experiment. The entire curve and full axis in 5B should be shown. In any event, this assay needs to be supplemented with direct NET quantification by microscopy.*

- We have performed new experiments that clearly demonstrate that NET formation following stimulation with NETs is increased in MICL-deficient neutrophils compared to WT neutrophils (new Figure 4B and Supplementary Figure 10E). In addition, we have included NET quantification by microscopy through the analysis of the MFI of cit-H3 and MPO (Supplementary Figure 10B).

10) *To accompany the PAD4 and DPI inhibition experiments, the authors should also provide microscopy images of NET-triggered NETs stained for NE, MPO, DNA and cit-H3 antibodies by microscopy and WB, to show that NETs trigger chromatin citrullination.*

- See comment above.

11) *Sup. Fig. 10A and 5A. Unstimulated controls should be shown to ascertain the degree to which WT neutrophils are stimulated to produce ROS by NETs. In Fig. S10A it appears that WT neutrophils do not produce any ROS, but in Fig. 5A there is some ROS production by WT cells. Adding unstimulated cells will help ascertain whether WT neutrophils respond at all to NETs in terms of ROS production, or whether this is only observed exclusively in KO cells.*

- We have included unstimulated cells in both Figure 4A and (new) Supplementary Figure 10C and show that WT neutrophils also produce ROS in response to NETs, but to a lesser extent compared to MICL-deficient cells (see comment above).

12) *Fig.5 and S10. Related to the previous point, there is a large discrepancy between the ROS production and NETosis assays. NETs barely activate ROS production in WT cells, but the difference in NET formation (by Sytox bulk fluorescence) is very small (10%). Thus, if WT neutrophils barely respond to NETs ROS-wise, and DPI blocks the process (Fig. S10B) why are the WT cells making 90% of the NETs that a KO neutrophil is making in response to NETs? I suspect that this is further evidence that the Sytox fluorescence assay is not the proper approach here.*

- We understand the reviewer's query regarding the discrepancy between the ROS production and NET formation assays. In light of the above comment, we have now included additional clearer data, which show that WT cells also produce ROS and NETs in response to NETs, but both processes are increased in MICL-deficient cells. In addition, we have now included microscopy-based quantification of NET formation by NETs.

13) *The lack of Proteinase K effect suggests that the interaction depends on DNA but not protein (Fig. 5D). Thus, MICL could be a DNA sensor and this could be tested with genomic neutrophil DNA alone or perhaps DNA oxidised in vitro. Moreover, genomic DNA does not trigger NETosis, but ROS stimulation is dependent on both DNA and protein (Fig. S10C). Falling short of dissecting the molecular determinants of NET-mediated stimulation, the authors could take advantage of these differences to further dissect the two activities. If MICL binds naked DNA, then it would be interesting to explore whether MSU crystal-triggered NETosis is blocked by naked DNA titrations. Naked DNA could also block NET-triggered NETosis is added in excess.*

- As suggested by the reviewer, using an ELISA-based assay we found that Fc-MICL binds gDNA and this interaction is abolished in the presence of antibodies targeting MICL (new Supp Fig 10H).

14) *Neutrophil density experiments might be interesting here. Increasing neutrophil confluence inhibits NET formation in response to PMA (unpublished observations). So, one could imagine that in the MSU or even PMA stimulation experiments, MICL KO neutrophils*

would be less sensitive to this unexplained density suppressing phenomenon. I don't think this is an essential experiment but it would support an important concept of NET density sensing which lingers below the surface in the manuscript.

- We thank the reviewer for this suggestion. We have performed this experiment using PMA as NET-inducer and we have not found any difference in NET formation between M1CL KO and WT neutrophils using four different cell density (5×10^4 , 1×10^5 , 2×10^5 and 4×10^5 cells/well).

For reviewer: NET release (quantified as percentage of Sytox green positive cells extruding NETs) in WT and M1CL-deficient neutrophils at different cell densities.

15) WB for cit-H3 and cell free DNA measurements would be helpful in the peritoneal inflammation model.

- We have performed cell free DNA measurements following NETs injection and found a trend towards higher levels of DNA in M1CL-deficient mice (Supplementary Figure 10I).

16) Fig. 6A It is strange that murine WT neutrophils do not make any "NETs" in response to *A. fumigatus*. (Again there is the issue that Sytox fluorescence readers do not distinguish between NETosis and necrosis, without microscopy data). The peak of NETosis as suggested by the fluorescence intensity in the M1CL KO is consistent with what should be expected for NETosis by WT neutrophils peaking between 4-8 hrs for hyphae. This is best visualized by video timelapse, but stills at 7-8 hrs will also be useful.

- We thank the reviewer for this suggestion and apologies for the confusion. WT neutrophils do produce NETs. We have added new data showing that WT neutrophils do release NETs in response to *A. fumigatus*, but this is increased in the absence of M1CL. We also have included representative microscopy images of neutrophils stimulated with *A. fumigatus* (Supplementary Figure 11A).

17) Fig. 6B and C. These experiments should be supported with cfu (kidney and spleen), NET stains (kidney) and systemic cytokines from day 2, rather than relying on indirect inferences

from PAD4 inhibitor treated mice to understand the mechanism of MICL protection. These are important readouts because the authors state that:

*“we confirmed that the increased survival of MICL-deficient mice was associated with higher NET formation, since the treatment of these animals with a PAD4 inhibitor increased the susceptibility to *A.fumigatus* infection to wild type levels (Fig. 6C).”line 240*

This interpretation of this experiment is oversimplified in the text because the mice in 6C should also be compared to their respective vehicle. Depending on variability across experiments, it appears that when one compares across 6B and 6C, PAD4-inh has opposing effects on WT and MICL KO mice, with WT being more protected and MICL KO being more susceptible upon PAD4 inhibition. If differences between the inhibitor and vehicle prove significant within each genotype, this would imply that the mechanisms of pathology also vary. So having the CFUs and NET stains, or cit-H3 WB of kidney lysates, and circulating un-citrullinated histones along with cytokines, IL-6, IL-1, IL-1RA (PMID:37059107) and G-CSF (PMID: 35945238) can help understand how NETosis, the cytokine storm and fungal control are affected by MICL.

- As suggested by the reviewer, we have now included CFU from brain, kidney and liver (Figure 5C and Supplementary Figure 11B), as well as systemic cytokines at day 2 post infection (Figure 5D). We found a significant reduction in fungal burdens in the brain in MICL-deficient animals. Serum levels of IL-6 and brain levels of G-CSF were significantly higher in infected WT compared to MICL-deficient animals (Fig. 5D and E), suggesting that MICL regulates systemic inflammation during invasive aspergillosis.
- To demonstrate that treatment with PAD4 inhibitor only affects MICL KO animals, we have included survival curves (and fungal burdens) of *Aspergillus*-infected WT and MICL KO mice treated with the inhibitor or vehicle control (Figure 6F and Supplementary Figure 11C). Thus, blocking NET formation increases susceptibility to *Aspergillus* infection of MICL-KO animals.

18) fig. 6A The ficoll gradient employed to isolate human neutrophils is notorious for pre-stimulating the cells. This may explain why the isotype in human cells is much more reactive than WT murine neutrophils. This can be determined with unstimulated controls which are absolutely necessary here, and the human data could be vastly improved if Histopaque/Percoll is used instead.

- We have now utilized unstimulated controls, as suggested.

Referee #2 (Remarks to the Author):

This study demonstrates that MICL is a regulator of NET release from neutrophils. A series of experiments show that in an arthritis model there are more neutrophils recruited and there is more injury in the MICL deficient mice. The work shows an association between NETs,

tissue injury and MICL antibodies in patients. The work also suggests that NETs bind MICL and that in a fungal infection not having MICL is of benefit.

There are two main issues that would need to be completely addressed in this paper to be acceptable. First, the mechanism by which MICL leads to less NET production is missing. While the authors show lots of increased neutrophil activation and they show more NET release the mechanism by which MICL regulates this is unclear. The authors show that MICL binds the DNA, but that is where the story ends. It should start there.

The second issue is related to the first and it is that the authors have published a preliminary study showing that arthritis is worse in MICL deficient mice, that this is related to myeloid cells and that arthritis patients have increased MICL antibodies. As such without addressing the first point and finding significantly more mechanistic results the work is only incremental based on their previous work.

- We have included new data revealing underlying mechanisms and new data showing that anti-MICL autoantibodies are also present in patients with severe COVID-19 and in patients with lupus (both NETs-mediated inflammatory diseases). Moreover, we show that these autoantibodies have functional and specific impacts on neutrophil functions.

Specific

There are additional important issues including what does the antibody to MICL do, does it activate the MICL to reduce neutrophil activation or does it block binding of DNA. This is not well developed.

- We have now included new data showing that MICL recognises DNA and that the antibody targeting MICL blocks binding of NETs/DNA to the receptor (Supplementary Figure 10F). The antibody does not activate MICL.

I also found it difficult to understand how the neutrophils were having similar amounts of chemokine receptors and other molecules and yet were recruited in greater numbers in MICL mice. A time course of neutrophil recruitment and NET production would address if indeed there is neutrophil recruitment that leads to NET release which recruits more neutrophils over time to lead to more NET release without MICL. The authors claim MICL regulates NET release presumably turning off this potent mechanism of action but with a single time point it is hard to understand how this conclusion can be made.

- To address this specific point, we now include new data showing neutrophil recruitment at the early stages of inflammation during CAIA and found that the number of neutrophils recruited to the ankle joints was higher in MICL-deficient animals at day 5, where there is no difference in the clinical score between WT and KO animals.

Referee #3 (Remarks to the Author):

This is an interesting paper suggesting that CLEC12A is a receptor for NETS and regulate arthritis. There are however some flaws in experimental settings and description of results.

1) The genetic background and experimental details of the MICL^{-/-} are not clear. Is this mouse on the same background as the used C57Bl mouse? Are they made in a B6 ES cell or with crisp? Are the experimental animals littermates? Are the experiments performed with the groups mixed in cages? Do the experiments follow the ARRIVE rules or something similar?

- MICL knockout mice (Clec12A^{-/-}) on a C57BL/6 background were produced by Taconic Artemis, USA as described in PMID: 26275430, and were bred and maintained under SPF conditions at the University of Aberdeen, University of Exeter and Charles River Laboratories.
- The arthritis models included in the manuscript were performed in males, and therefore, MICL KO and WT animals were housed in different cages. To confirm that the phenotype is maintained when both genotypes are co-housed, we performed the CAIA model in co-housed females, and found higher levels of inflammation in MICL KO animals. Thus co-housing does not impact our results. Our experiments do follow ARRIVE guidelines (see M&M).

For reviewer: CAIA clinical severity scoring over time in WT and MICL-deficient females co-housed (n=five animals/group).

2) Its not clear whether the observed expansion and activation of PMN in the joints were the results of a direct effect by the absence of Clec12a or if it was secondary to the arthritis per se. In Fig 1 the data are based on three different experiments, is it only one selected experiment shown or are the data normalized and shown for all used mice? Is the data is selected all data need to be shown, either normalized or separate, how can we otherwise know which experiment is selected? It seems that similar problems are in all showed in vivo experiments. Apparently the increased arthritis susceptibility of the MICL mice has been previously reported – so, what is new? Its not very surprising that there are more activated PMNs in an exaggerated CAIA arthritis.

- As detailed in the response to reviewer 2 above, it is the mechanism underlying the increased arthritic inflammation that is the fundamental discovery here, in that we have discovered that MICL directly recognises, as well as regulates formation of, neutrophil extracellular traps. Moreover, in our revised manuscript we provide new data showing the underlying mechanism is also relevant to other NET-related autoimmune/inflammatory diseases including LUPUS and severe COVID 19.

- We now include neutrophil recruitment at the early stages of inflammation during CAIA and found that the number of neutrophils recruited to the ankle joints was higher in MICL-deficient animals at day 5, where there is no difference in the clinical score between WT and KO animals. This shows that the observed expansion of neutrophils in the joints of MICL-deficient animals is not secondary to higher levels of arthritis. We have modified the text to better reflect this.
- As requested by the reviewer we have now pooled the data from at least three independent experiments or state the number of animals and repeats of all the results showed in the manuscript.

3) The transfer of BM PMN is novel and indicate that the Clec12a effect is restricted to PMN. However, it's not clear how pure the transferred population is. PMN are sensitive cells whereas other BM cells are not, it's difficult (but important) to exclude other BM cells such as macrophages in these experiments. If there are 10% other cells, monocytes/macrophages etc, these could explain the effect.

- We have now included an example of the gating strategy used to identify the neutrophil population post purification from bone marrow.

4) MSU, but not PMA induced enhanced ROS in clec12a deficient PMN. Not surprisingly it induces more NET formation. An increased NET formation was seen in arthritis joints (again not so surprising as this is expected because of more PMN in an arthritic joint). Surprisingly PAD4 inhibitors blocked the development of NETs (and arthritis) only in MICL knockout mice and not in wildtype? Why not in wildtype? Also wt has NETS.

- As detailed above, the novelty of our discovery is the mechanism underlying the development of these responses. We have modified the text to reflect this point more clearly.
- As the reviewer points out wild type mice do induce NETs, but not through a PAD4 pathway. This has been shown both for arthritis (PMID: 22551352, 21346230) and in models of systemic aspergillosis (PMID: 29896200). Thus, our data indicate that MICL plays a fundamental role in suppressing this pathway in these models in mice, which we have better highlighted in the discussion.

5) It's not correct to state that CIA gives low penetrance in B6 mice. In fact, it is not possible to induce CIA in B6 mice. The observed arthritis in B6 mice after immunization with collagen is due to a contamination. There is a however a problem with DBA/1 as well since the authors regroup the mice after puberty, this will commonly induce a spontaneous form of arthritis or enhanced CIA, which is due to other factors than collagen autoimmunity. The experimental settings need to be different to avoid this actually well described phenomena.

- We have modified our comments regarding penetrance of CIA in C57BL/6 mice.
- We understand the reviewer concern about regrouping DBA/1 animals since previous studies have shown that spontaneous form of arthritis develop from the age of 12 weeks onwards (PMID: 23253472, 15082495). However, we had tested the impact of regrouping DBA/1 mice in our animal facility, and found no naïve animal developed any spontaneous form of arthritis by the age of 13 weeks (the latest time point in our experiments). We have included a comment to this effect in our manuscript. Notably,

animals were randomly allocated to different treatments: anti-MICL antibody, isotype control or PBS, and as we had seen in the other models of arthritis, only administration of anti-MICL antibodies exacerbated CIA disease in DBA/1 mice.

6) *The statement that antibodies to Clec12a correlates with RA is not well funded. To me it seems to be part of a known increase of natural autoantibodies in RA, since it correlates well with RF.*

- We respectfully disagree with the reviewer. The key correlation is that anti-MICL antibodies significantly correlate with anti-CCP antibody levels. Anti-CCP levels are one marker indicating severity of disease (PMID: 15621577, 16176994, 16207336), but importantly are also an indicator of NETosis (PMID: 28182854).

7) *The definition of Clec12a as a receptor for NETS, or rather DNA within NETs is maybe not that surprising but is likely to lack specificity. Many CLRs, and other proteins, do the same. Controls for specificity is lacking.*

- This is the first mammalian C-type lectin receptor (and the first inhibitory receptor) ever described to recognise DNA. However, as requested by the reviewer, we have now included appropriate C-type lectin controls in all relevant experiments.

8) *It is well know that a major regulatory gene in autoimmune disease and RA, as well as its mouse models, is Ncf1. Mutations in Ncf1 leads to increased arthritis, decreased ROS and lack of NETs. If Clec12a is uniquely promoting arthritis through NETs it should be possible to do an experiment with Ncf1 mutated mice i.e crossing the MICL knockout with the Ncf1 mutation and investigate the littermates. Or preferably use LysG promotor conditioned MICL knockout (or conditioned knockin on MICL knockout background) mice.*

- We thank the reviewer for this suggestion, but feel that including NCF1 is beyond the scope of the current work. As detailed above our primary discovery is that MICL recognises and regulates NETs, and provide evidence for its relevance beyond arthritis, including LUPUS and COVID-19, and in response to fungal infection.

Reviewer Reports on the First Revision:

Referees' comments:

Referee #1 (Remarks to the Author):

The authors have addressed the majority of my comments and the new data are more robust and convincing.

One exception is comment 18 where the data in Fig. 5a still do not include unactivated controls for mouse and human cells, only activated WT and isotype controls respectively. I suspect the authors forgot to include these, and it is mostly a trivial issue that can be addressed without re-review.

A second minor point is that it is difficult to ascertain a difference in NETosis in response to *A. fumigatus* in KO neutrophils (Suppl. fig. 11), but admittedly a 25% difference is not easy to notice by eye and can only be reflected in digital quantification on multiple images which is already provided.

Referee #2 (Remarks to the Author):

Excellent rebuttal.

Author Rebuttals to First Revision:

Response to Reviewers comments:

We thank again the Referees for the evaluation of our manuscript.

Referee #1 (Remarks to the Author):

The authors have addressed the majority of my comments and the new data are more robust and convincing.

One exception is comment 18 where the data in Fig. 5a still do not include unactivated controls for mouse and human cells, only activated WT and isotype controls respectively. I suspect the authors forgot to include these, and it is mostly a trivial issue that can be addressed without re-review.

- We apologize for this omission. We have now included unactivated controls for mouse and human cells in Fig 5a.

*A second minor point is that it is difficult to ascertain a difference in NETosis in response to *A. fumigatus* in KO neutrophils (Suppl. fig. 11), but admittedly a 25% difference is not easy to notice by eye and can only be reflected in digital quantification on multiple images which is already provided.*

- As the reviewer points out, we have performed digital quantification on multiple images and this is already included as part of the figure.

Referee #2 (Remarks to the Author):

Excellent rebuttal.